# Epithelial SIRT6 governs IL-17A pathogenicity and drives allergic airway inflammation and remodeling

Jingyun Quan[1,2,6], Xiaoxia Wen[3,6], Guomei Su[3,6], Yu Zhong[3,6], Tong Huang[3], Zhilin Xiong[3], Jiewen Huang[3], Yingying Lv[1], Shihai Li[3], Shuhua Luo[4], Chaole Luo[1], Xin Cai[3], Xianwen Lai[3], Yuanyuan Xiang[3], Song Guo Zheng [5], Yiming Shao[1], Haitao Lin[2], Xiao Gao[3] ✉, Jing Tang [4] ✉ & Tianwen Lai[1,3] ✉

Dysregulation of IL-17A is closely associated with airway inflammation and remodeling in severe asthma. However, the molecular mechanisms by which IL-17A is regulated remain unclear. Here we identify epithelial sirtuin 6 (SIRT6) as an epigenetic regulator that governs IL-17A pathogenicity in severe asthma. Mice with airway epithelial cell-specific deletion of *Sirt6* are protected against allergen-induced airway inflammation and remodeling via inhibiting IL-17A-mediated inflammatory chemokines and mesenchymal reprogramming. Mechanistically, SIRT6 directly interacts with RORγt and mediates RORγt deacetylation at lysine 192 via its PPXY motifs. SIRT6 promotes RORγt recruitment to the IL-17A gene promoter and enhances its transcription. In severe asthma patients, high expression of SIRT6 positively correlates with airway remodeling and disease severity. SIRT6 inhibitor (OSS_128167) treatment significantly attenuates airway inflammation and remodeling in mice. Collectively, these results uncover a function for SIRT6 in regulating IL-17A pathogenicity in severe asthma, implicating SIRT6 as a potential therapeutic target for severe asthma.

Asthma is a chronic inflammatory disease of the airways affecting >300 million patients worldwide, which severe asthma patients representing <10% but are responsible for the most share of clinical cost and mortality[1–3]. Chronic airway inflammation and airway remodeling refers to bronchial structural changes and is a critical pathologic feature of severe asthma. The extent of airway remodeling is strongly associated with lung function reduction and steroid resistance[2]. However, the mechanisms underlying the airway inflammation and remodeling process in severe asthma are not completely understood.

Severe asthma patients have eosinophilic inflammation or neutrophilic inflammation or mixed granulocytic (both eosinophilic and neutrophilic inflammation). Common indoor and outdoor environmental exposures can influence airway inflammation in asthma[1]. Airway epithelial cells, the first site of environmental factors exposure such as house dust mites (HDM), pollen, and lipopolysaccharide (LPS), conduct airways develop chronic inflammation, and ultimately lead to airway remodeling[4]. HDM is a complex mixture, containing mite allergens and microbial products. LPS is an endotoxin derived from the membrane of colonizing Gram-negative bacteria and environmental contamination,

[1]Department of Respiratory and Critical Care Medicine, The First Dongguan Affiliated Hospital, Guangdong Medical University, Dongguan 523710, China. [2]Department of Health Management & Physical Examination Center, Affiliated Hospital of Guangdong Medical University, Zhanjiang 524001, China. [3]Institute of Respiratory Diseases, Affiliated Hospital of Guangdong Medical University, Zhanjiang 524001, China. [4]Department of Anesthesiology, Affiliated Hospital of Guangdong Medical University, Zhanjiang 524001, China. [5]Dongguan Key Laboratory of Chronic Inflammatory Diseases, The First Dongguan Affiliated Hospital, Guangdong Medical University, Dongguan 523710, China. [6]These authors contributed equally: Jingyun Quan, Xiaoxia Wen, Guomei Su, Yu Zhong. ✉e-mail: gaoxiao7187676@126.com; tanglitangjing@126.com; laitianwen2011@163.com

and the level of LPS is correlated with the severity of asthma and decline in lung function[4]. The pathological process involves a combination of increased epithelial-mesenchymal transition (EMT), myofibroblast transition, and extracellular matrix (ECM) deposition[5]. Recent studies indicated a pivotal role for interleukin (IL)−17A in the EMT process and contributes to severe asthma[6,7]. Unlike other asthma phenotypes, the IL-17A-related neutrophilic asthma (NA) phenotype exhibits frequent asthma exacerbations or more severe asthma. IL-17A promotes airway goblet cell hyperplasia, mucus hypersecretion, and fibrosis, which contribute to airway remodeling[8]. However, how this cytokine promotes airway inflammation and remodeling, and accelerates disease progression in severe asthma remains unclear.

Acetylation is a widely occurring epigenetic modification of proteins that are involved in diverse biological processes[9,10]. The acetylation status is maintained by an intricate balance of histone deacetylase (HDAC) and histone acetyltransferase (HAT)[11,12]. In recent years, the physiological functions of the sirtuin deacetylase family (SIRT1-SIRT7) have been studied. Among the seven sirtuins, SIRT6 plays a leading role in regulating cell proliferation, stress response, genomic stability, energy balance, and aging[13,14]. However, whether and how SIRT6 regulates IL-17A pathogenicity in severe asthma has not been deciphered.

In the present study, using various human samples (eg, airway biopsy samples, bronchoalveolar lavage fluid [BALF], peripheral blood) and different mouse models of severe asthma, we identify epithelial SIRT6 as an important epigenetic regulator for IL-17A secretion during severe asthma progression. Deletion or inhibition of SIRT6 results in impaired IL-17A secretion in airway epithelium and ameliorates airway remodeling in severe asthma via RORγt-K192 deacetylation. Together, these data support the therapeutic hypothesis that blocking SIRT6 could prevent IL-17A-mediated airway inflammation and remodeling in severe asthma.

## Results

### Increased expression of SIRT6 in asthma correlates with disease severity

To assess the role of SIRT6 in the pathogenesis of asthmatic airway remodeling, we initially detected the expression of SIRT6 and other sirtuin deacetylases (SIRT1-SIRT5, SIRT7) in the lung tissues in an experimental model of acute severe asthma (ASA) (Fig. 1a). As assessed by quantitative RT-PCR (qRT-PCR), we observed that *Sirt6* was the most significant high expressed SIRT members compared with controls, whereas *Sirt1, Sirt3*, and *Sirt5* expression were lower in the lung tissues of asthmatic mice than in controls. We did not detect obvious changes in *Sirt2* and *Sirt4* mRNA expression in this model (Fig. 1b, c). Consistently, the protein abundance of SIRT6 was also significantly up-regulated, whereas SIRT1, SIRT3, and SIRT5 were down-regulated in the lung tissues of asthmatic mice compared to controls (Fig. 1d, e). We next examined whether SIRT6 is also elevated in asthmatic patients (Table S1). Consistent with animal data, *Sirt1, Sirt3*, and *Sirt5* expression was decreased in the peripheral blood of patients with asthma compared to controls (Fig. 1f). However, the expression of *Sirt6* mRNA and protein was significantly up-regulated in the peripheral blood and bronchial biopsies of patients with asthma compared with those seen in healthy controls. In the subgroup, the SIRT6 levels were higher in patients with severe asthma than those in control individuals and mild to moderate asthmatic patients (Fig. 1g-i, and Supplementary Fig. 1a, b).

By using Spearman correlation analysis, the *Sirt6* levels were correlated negatively with forced expiratory volume in 1 second (FEV₁) ($r = -0.41$, $P = 0.015$) and asthma control test (ACT) score ($r = -0.39$, $P = 0.023$) (Fig. 1j-m). We further found that the percentage of neutrophils were higher in severe asthma patients compared to the other two groups (Fig. 1n). Moreover, the *Sirt6* mRNA levels were positively correlated with neutrophils percentage ($r = 0.48$, $P = 0.005$) (Fig. 1o).

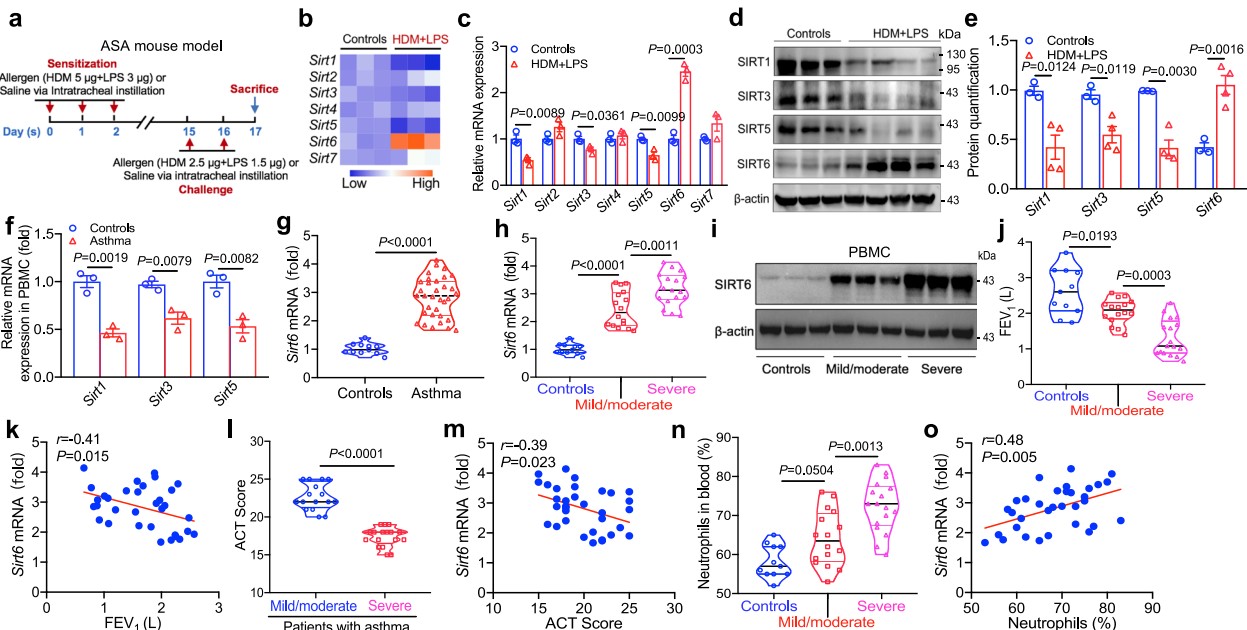

**Fig. 1 | Increased SIRT6 expression is positively related to asthma severity. a** Schematic illustrating an established HDM/LPS-induced acute severe asthma (ASA) mouse model. **b, c** qRT-PCR analysis of the expression of the SIRT family (left margin) in mouse lung tissues. Results were normalized to those of the gene encoding β-actin. **d, e** Western blot analysis of the SIRT family (SIRT1, SIRT3, SIRT5, and SIRT6). Quantification of SIRT6 expression was analyzed by using Image J software. **f, g** mRNA levels of *Sirt6* in peripheral blood from healthy participants ($n = 11$) and asthmatic patients ($n = 33$). **h, i** Asthmatic patients were divided into mild to moderate asthma ($n = 16$) and severe asthma ($n = 17$) according to their disease severity. mRNA and protein levels of *Sirt6* in peripheral blood were assessed. **j, k** Postbronchodilator forced expiratory volume in 1 second (FEV₁) was assessed. Correlation between *Sirt6* expression in peripheral blood and FEV₁ was determined by Spearman analysis. **l, m** Asthma control test (ACT) score was analyzed. Correlation between *Sirt6* expression in peripheral blood and ACT score was determined by Spearman analysis. **n, o** Percentage of neutrophils in peripheral blood was analyzed. Correlation between *Sirt6* expression and neutrophils percentage in peripheral blood was determined by Spearman analysis. Data are shown as means ± SEM and three or more independent experiments were performed. Significance was calculated by Two-tailed unpaired Student's *t*-test for (**a–f, g–l**); one-way ANOVA followed by Tukey's post-hoc test for (**h–j–n**).

Altogether, these data revealed that SIRT6 was increased during asthma and correlated with disease severity and may be a potential biomarker for severe asthma.

## SIRT6 is highly expressed in airway epithelium cells and promoted mesenchymal reprogramming

To define the cellular source of SIRT6 in the lung during severe asthma, we using published single-cell RNA-seq (scRNA-seq) data[15]. The analysis of database revealed that airway epithelial cells (Supplementary Fig. 2a, b) and other cells such as macrophages, different T cell populations have an expression of SIRT6 (Supplementary Fig. 2c, d), indicated that SIRT6 is expressed in a cell-nonspecific manner. We next examined the colocalization of SIRT6 with SCGB1A1 (airway epithelium cells marker), F4/80 (macrophage marker), CD31 (endothelial marker), and α-SMA (smooth muscle cells marker) in the lung tissues of asthmatic mice. SIRT6 was largely expressed in airway epithelium cells, whereas low levels of SIRT6 were observed in macrophages. SIRT6 was not observed in endothelial cells and smooth muscle cells (Fig. 2a). To confirm this finding, we used different human cell lines including human bronchial epithelium (HBE) cells, non-polarized macrophages

(M0) differentiated from THP-1, human vein endothelial cells (VEC), and human pulmonary artery smooth muscle cells (SMC), to measure the expression of SIRT6. Consistent with the above results, an abundant amount of SIRT6 was observed in HBE cells, whereas macrophages expressed a low level of SIRT6. SIRT6 was not observed in the other cells (VEC, SMC) (Fig. 2b).

We then focused on the role of airway epithelial SIRT6 in the pathogenesis of asthmatic airway remodeling. Using HBE cells, we further observed that SIRT6 mRNA and protein were significantly elevated in allergen-induced HBE cells (Fig. 2c, d). These results were further confirmed by immunofluorescence (IF) (Fig. 2e, f). Furthermore, SIRT6-positive cells in airway epithelium were significantly elevated from asthmatic mice compared with the control mice (Fig. 2g, h).

Following environmental insults, airway epithelium promotes the process of epithelial-mesenchymal transition (EMT) program and results in airway remodeling in asthma[16]. We next asked whether SIRT6 regulates the EMT process related to asthmatic airway remodeling. Allergic stimulation induced HBE cells to elongated shape and enhances actin filament formation (Fig. 2i), which are characteristic

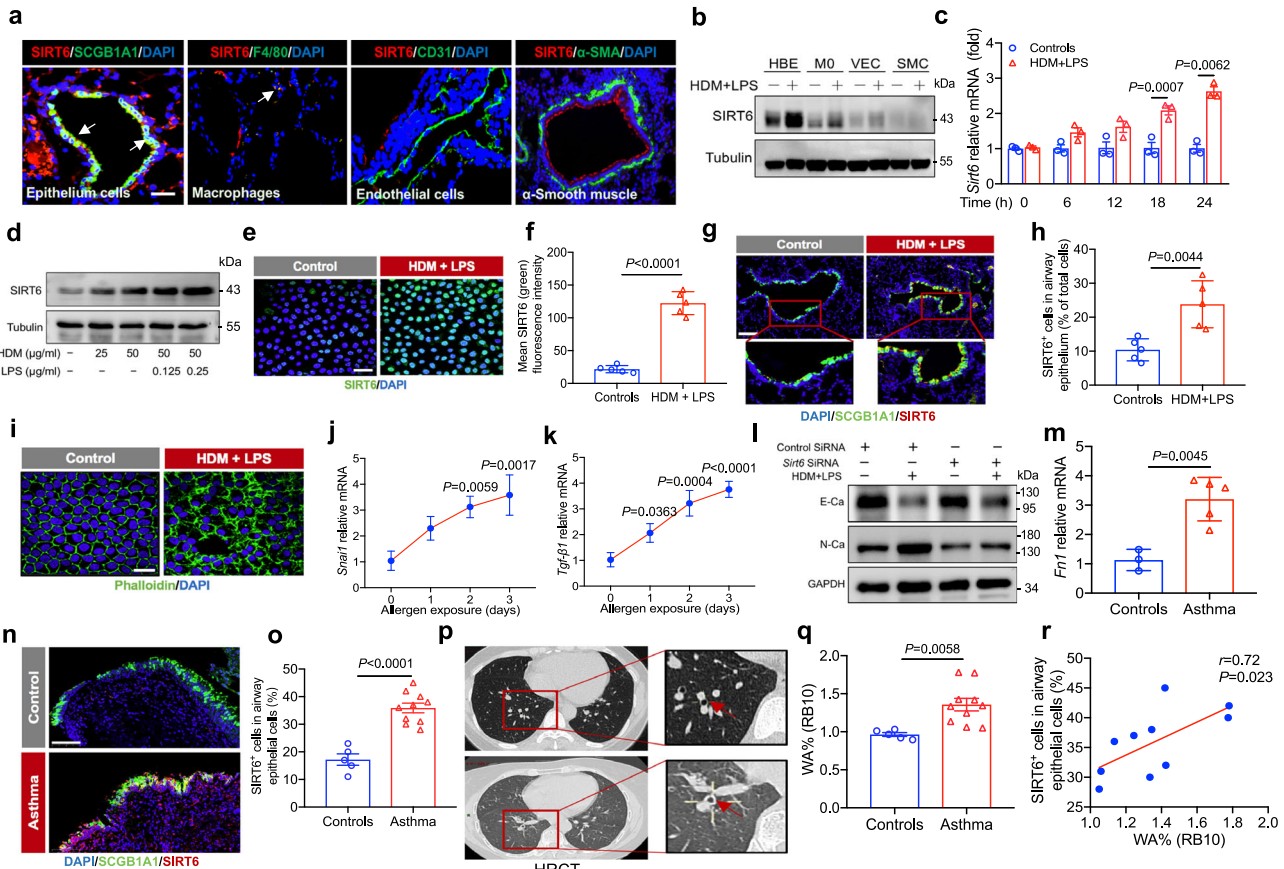

**Fig. 2 | SIRT6 was highly expressed in airway epithelium cells and promoted mesenchymal transition. a** IF staining for SIRT6 with SCGB1A1, F4/80, CD31, or α-SMA. Scale bars, 100 μm. **b** the expression of SIRT6 in HBE cells, M0, VEC, and SMC as determined by Western blot. **c, d** Sirt6 epxression in allergen-induced HBE cells for different times and different doses was analyzed. **e, f** Representative IF staining of SIRT6 in HBE cells stimulated with allergen (n = 5). Scale bars, 20 μm.
**g, h** Representative IF staining of SIRT6 (Red) in the airway epithelium (SCGB1A1, Green) from asthmatic and control mice (n = 5). Quantification was done using Image J software. Scale bars, 100 μm. **i** IF micrographs of HBE cells incubated in the absence or presence of HDM/LPS for 15 days. Cells were stained with Alexa Fluor 488-conjugated phalloidin (green color) and DAPI (blue color). Scale bars, 20 μm. **j, k** qRT-PCR for the EMT regulators Snai1, Tgf-β1 in HBE cells stimulated with HDM/LPS. **l** HBE cells were transfected with Sirt6 siRNA for 24 h and then treated with

HDM/LPS for another 48 h. The expression of E-Ca and N-Ca was measured using western blot. **m** qRT-PCR for bronchial mucosal biopsy specimens from healthy participants (n = 3) or patients with asthma (n = 5) for the EMT regulator Fn1 mRNA expression. **n, o** Representative of SIRT6 expression in bronchial biopsy specimens (Controls n = 5; Asthma n = 10). **p** Representative of high-resolution CT (HRCT) images of bronchial wall thickening in control participants (n = 5) and asthmatic patients (n = 10). **q** Single-slice airway measurements were collected in the apical bronchus of the right lower lobe (RB10) (Controls n = 5; Asthma n = 10).
**r** Correlation between SIRT6 expression in airway epithelium and airway wall thickness in asthmatic patients was investigated using Spearman analysis. Data are shown as means ± SEM and three or more independent experiments were performed. Significance was calculated by Two-tailed unpaired Student's t-test for (**c-f-h-m-o-q**); one-way ANOVA followed by Tukey's post-hoc test for (**j, k**).

factors of the EMT program. Allergen exposure also induced the levels of key remodeling markers, such as N-Ca, vimentin, snail family transcriptional repressor 1 (SNAI1), and TGF-β1 (Fig. 2j, k and Supplementary Fig. 3a-d). Moreover, SIRT6 knockdown reduced the allergen–induced production of N-Ca, but increased E-Ca expression (Fig. 2l and Supplementary Fig. 3e, f). We also found that the EMT core regulator such as fibronectin (FN1), SNAI1, and programmed cell death ligand 1 (PD-L1) in the mucosa of asthmatic patients were significantly increased compared with controls (Fig. 2m, Supplementary Fig. 3g-i, and Table S2), suggesting that activation of the EMT process in the airway epithelium of patients with asthma. To further determine whether SIRT6 expression in the airway epithelium and is relative to airway remodeling in asthmatic patients, SIRT6 expression in bronchial biopsy specimens was evaluated using IF staining and airway wall thickening was analyzed by using high resolution CT (HRCT) technique. Compared with the control participants, SIRT6 positive cells were significantly elevated in the airway epithelium of bronchial biopsy samples from asthmatic patients (Fig. 2n, o). Bronchial wall thickening was obviously elevated in asthmatic patients (Fig. 2p, q). Moreover, the expression of SIRT6 in the airway epithelium showed a significant positive correlation with bronchial wall thickening ($r = 0.72$, $P = 0.023$) (Fig. 2r).

Collectively, these data revealed that SIRT6 was highly expressed in airway epithelium cells and potentially promoted asthmatic airway remodeling, which also prompted us to further explore the precise role of epithelial SIRT6 in the airway remodeling process in severe asthma.

## SIRT6 deficiency attenuates airway inflammation and remodeling in asthma

To dissect how SIRT6 exerts its function in severe asthma, we generated mice with airway epithelium cell-conditional knockout of *Sirt6* (*AE-Sirt6*$^{\Delta/\Delta}$). All mice were genotyped with PCR (Fig. 3a, b). The absence of SIRT6 from airway epithelium cells was verified by Western blot, qRT-PCR, and IF (Supplementary Fig. 4a, b). SIRT6 deficiency in *AE-Sirt6*$^{\Delta/\Delta}$ mice did not influence the development of immune cells, such as CD4$^+$ T cells and macrophages (Supplementary Fig. 4c-g and Supplementary Fig. 5). Control (*AE-Sirt6*$^{fl/fl}$) and *AE-Sirt6*$^{\Delta/\Delta}$ mice were exposed to HDM/LPS according to a chronic severe asthma (CSA) model as described in Supplementary Fig. 4h, which associated airway remodeling[17,18]. Interestingly, we detected that several features of airway remodeling examined by peribronchial trichrome (Masson) staining, periodic acid Schiff (PAS) staining, and α-smooth muscle actin (α-SMA) were reduced in HDM/LPS-exposed *AE-Sirt6*$^{\Delta/\Delta}$ mice (Fig. 3c-f and Supplementary Fig. 4i). The EMT relative markers such as N-Ca, and SNAI1 were also significantly decreased in *AE-Sirt6*$^{\Delta/\Delta}$ mice compared with *AE-Sirt6*$^{fl/fl}$ mice exposed to HDM/LPS (Fig. 3g-i).

Persistent airway inflammation results in epithelial damage and EMT which ultimately lead to airway remodeling[19]. We further investigate whether SIRT6 regulates the EMT process in severe asthma by modulating airway inflammation. The total inflammatory cells, neutrophils, eosinophils, and lymphocytes in BALF were reduced in HDM/LPS-exposed *AE-Sirt6*$^{\Delta/\Delta}$ mice (Fig. 3j, k). Hematoxylin & eosin (HE) staining and quantitative analysis showed that SIRT6 deficiency attenuated allergen-induced peribronchial infiltrates of inflammatory cells

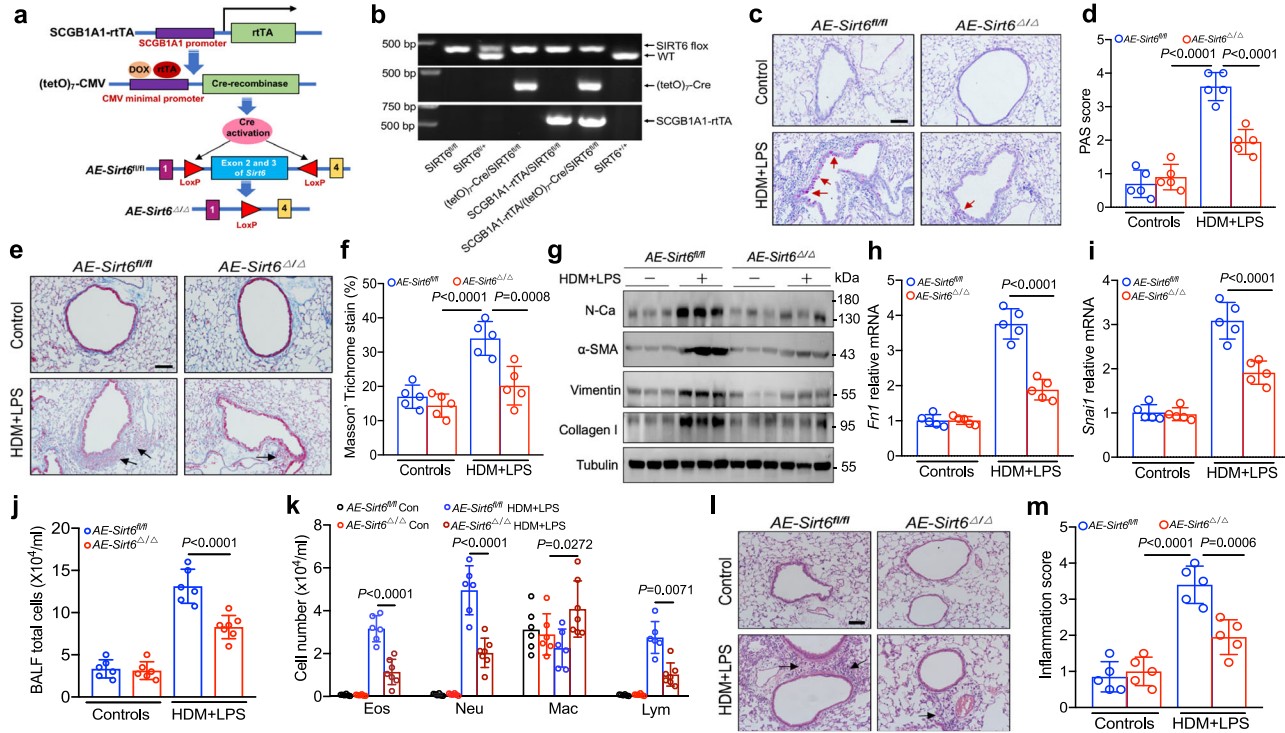

**Fig. 3 | SIRT6 deficiency protects against airway remodeling in mouse model of asthma. a** Schematic illustrating the genetic approach used to generate airway epithelium cell–conditional knockout of *Sirt6* (*AE-Sirt6*$^{\Delta/\Delta}$) mice. **b** SIRT6 deficiency was confirmed by assessing genomic DNA. **c-f** Representative periodic acid-Schiff (PAS) staining and Masson of lung sections from *AE-Sirt6*$^{fl/fl}$ and *AE-Sirt6*$^{\Delta/\Delta}$ mice treated with HDM/LPS in a chronic severe asthma (CSA) model and their quantification (*n* = 5 per group per study). Scale bars, 100 μm. **g** Western blot analysis of the EMT markers N-Ca, vimentin or the myofibroblast markers α-SMA and Collagen I in lung homogenate from *AE-Sirt6*$^{fl/fl}$ and *AE-Sirt6*$^{\Delta/\Delta}$ mice treated with HDM/LPS in CSA model (*n* = 5 per group per study). **h, i** qRT-PCR analysis of the EMT regulators *Fn1*

and *Snai1* in lung homogenate from *AE-Sirt6*$^{fl/fl}$ and *AE-Sirt6*$^{\Delta/\Delta}$ mice treated with HDM/LPS in CSA model (*n* = 5 per group per study). **j-m** Total BAL fluid (BALF) cells, differential cell counts, and histologic analysis of the lung sections were performed with hematoxylin and eosin staining to visualize inflammatory cell recruitment from *AE-Sirt6*$^{fl/fl}$ and *AE-Sirt6*$^{\Delta/\Delta}$ mice treated with HDM/LPS in ASA model (*AE-Sirt6*$^{fl/fl}$-Control, *AE-Sirt6*$^{\Delta/\Delta}$-Control, *AE-Sirt6*$^{fl/fl}$-Asthma *n* = 6; *AE-Sirt6*$^{\Delta/\Delta}$-Asthma *n* = 7). Data are shown as means ± SEM and three or more independent experiments were performed. Significance was calculated by one-way ANOVA followed by Tukey's post-hoc test for (**d-f-h-k-m**).

(Fig. 3l, m). Together, these data indicated that SIRT6 deficiency was important for protecting against airway inflammation and remodeling in severe asthma via the downregulation of mesenchymal reprogramming and airway inflammation.

### SIRT6 deficiency inhibits epithelial IL-17A secretion in vivo and in vitro

To investigated the potential mechanisms of SIRT6 in airway remodeling in severe asthma, we first performed RNA sequencing (RNA-seq) for the airway tissues from wild-type (WT) mice exposed to normal saline or HDM/LPS. There were down-regulation of 214 genes and up-regulation of 456 genes (Supplementary Fig. 6a and Supplementary Data 1). These genes were mainly categorized into primary immunodeficiency, asthma, and NF-kappa B signaling pathway according to the Kyoto Encyclopedia of Genes and Genomes (KEGG) pathway enrichment analysis (Fig. 4a). Gene Ontology (GO) clustering of biological processes identified that response to stimulus, metabolic process, and immune system process were significantly enriched in the biological

processes (Supplementary Fig. 6b). Th17-type immune response and IL-17A expression, which play a critical role in airway remodeling[8], were significantly upregulated in asthma mice compared with control mice (Fig. 4b, c). Further, we dertermine whether SIRT6 regulates epithelial IL-17A expression in asthmatic airway remodeling using $Sirt6^{fl/fl}$ mice and $AE\text{-}Sirt6^{\Delta/\Delta}$ mice. We observed that HDM/LPS exposure increased the expression of IL-17A mRNA and protein in the airway tissues of $Sirt6^{fl/fl}$ mice, whereas significantly decreased in $AE\text{-}Sirt6^{\Delta/\Delta}$ mice (Fig. 4d-g). Using IF staining, IL-17A expression in airway epithelial cells was reduced in $AE\text{-}Sirt6^{\Delta/\Delta}$ mice compared to the $Sirt6^{fl/fl}$ mice exposed to HDM/LPS (Fig. 4h, i). Consistent with animal model data, airway epithelial IL-17A expression was obviously up-regulated in asthmatic patients compared with the control participants (Fig. 4j, k). Moreover, the amount of IL-17A secreted into the BALF was a positive correlation with the expression of SIRT6 in the airway epithelium (Fig. 4l, m). In vitro, HDM/LPS exposure obviously promoted the expression of IL-17A, but SIRT6 small interfering RNA (siRNA) decreased IL-17A level in comparison with the HBE cells receiving control siRNA (Fig. 4n, o and

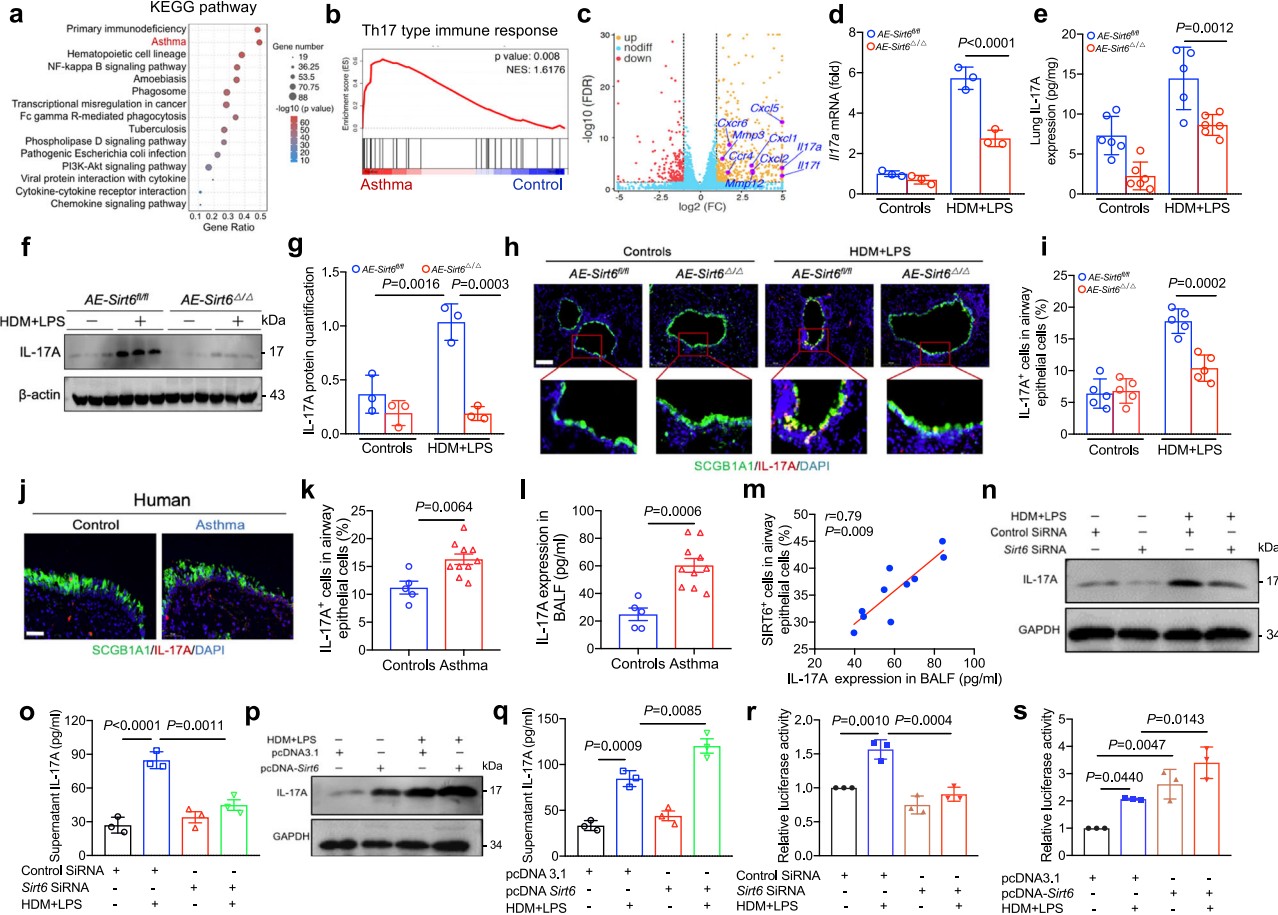

**Fig. 4 | SIRT6 deficiency inhibits IL-17A expression in airway epithelium.**
**a** KEGG enrichment analysis in the lung tissues of WT mice in ASA model compared to control. **b** Gene expression of Th17 type immune response was assessed by Gene set enrichment analysis (GSEA). **c** A volcano plot showed Il-17A and other upregulated genes encoding inflammatory products (red, downregulated genes; brown, downregulated genes). **d**−**g** qRT-PCR ($n = 3$), ELISA ($AE\text{-}Sirt6^{fl/fl}$-Control, $AE\text{-}Sirt6^{\Delta/\Delta}$-Control, $AE\text{-}Sirt6^{\Delta/\Delta}$-Asthma $n = 6$; $AE\text{-}Sirt6^{fl/fl}$-Asthma $n = 5$) and Western blot ($n = 3$) analysis of IL-17A in lung homogenates from $AE\text{-}Sirt6^{fl/fl}$ and $AE\text{-}Sirt6^{\Delta/\Delta}$ mice treated with HDM/LPS in ASA model. **h, i** Representative of IL-17A expression (Red) in airway epithelium cells (SCGB1A1, Green) of $AE\text{-}Sirt6^{fl/fl}$ and $AE\text{-}Sirt6^{\Delta/\Delta}$ mice treated with HDM/LPS in ASA model. Quantification was done using Image J software ($n = 5$). **j, k** Representative of IL-17A expression (Red) in airway epithelium cells (SCGB1A1, Green) of bronchial biopsy samples from control donors ($n = 5$) and

asthmatic patients ($n = 10$). Quantification was analyzed by using Image J software. **l, m** The levels of IL-17A in the BALF from control donors ($n = 5$) and asthmatic patients ($n = 10$) were detected using ELISA. The correlation between SIRT6 expression in airway epithelium and IL-17A levels in BALF was investigated using Spearman analysis. **n−q** HBE cells were transfected with $Sirt6$ small interfering RNA (siRNA) or $Sirt6$ plasmid for 24 h and were then treated with HDM/LPS for another 24 h. The expression of IL-17A was studied by using Western blot and ELISA analysis. **r, s** HBE cells were pretreated with $Sirt6$ siRNA or $Sirt6$ plasmid for 24 h and then transfected with the IL17A-reporter plasmid. Luciferase activity was then measured in HDM/LPS-treated and untreated cells. Data are shown as means ± SEM and three or more independent experiments were performed. Significance was calculated by one-way ANOVA followed by Tukey's post-hoc test for (**d, e−g−i−k, l−o−q, r, s**).

Supplementary Fig. 6c-e). Conversely, overexpression of SIRT6 promoted HDM/LPS-induced IL-17A expression (Fig. 4p, q and Supplementary Fig. 6e). To further explore whether SIRT6 interacts with IL-17A promoter in the airway epithelium, we used HBE cells to perform an IL-17A promoter-driven luciferase reporter assay. As shown in Fig. 4r, the up-regulation of IL-17A promoter activity by allergen was blocked by *Sirt6* siRNA. In contrast, the luciferase activity of IL-17A was increased by the overexpression of SIRT6 (Fig. 4s).

Previous studies have shown that IL-17A can induce the production of matrix metalloproteases (MMPs) and cytokines, which have an important role in the pathogenesis of airway remodeling[19–21]. The RNA-seq data also showed that *Il-17f*, *Mmp3*, *Mmp12*, *Cxcl1*, and *Cxcl2* were increased in the asthmatic mice (Fig. 4c). To confirm this result, qRT-PCR and ELISA were performed. The expression of MMPs and cytokines such as *Mmp3*, *Mmp12*, and *Cxcl1* were much lower in the airway tissues of *AE-Sirt6*[Δ/Δ] mice than in that of *AE-Sirt6*[fl/fl] mice exposed to HDM/LPS (Supplementary Fig. 6f-i). Moreover, the levels of eosinophilic inflammation (IL-5, IL-13) and neutrophilic inflammation (KC) in the BALF of *AE-Sirt6*[Δ/Δ] mice than in that of *AE-Sirt6*[fl/fl] mice exposed to HDM/LPS (Supplementary Fig. 6j-l).

Collectively, these results indicated that elevated epithelial SIRT6 promoted IL-17A release, which induced MMPs and cytokines (eg, *Mmp3*, *Mmp12*, and *Cxcl1*) scretion in severe asthma.

### SIRT6 deficiency attenuates airway remodeling via an IL-17-dependent mechanism

To assess whether SIRT6 deficiency attenuates airway remodeling through an IL-17A-dependent mechanism. We first investigated whether exogenous IL-17A is able to aggravate EMT in HBE cells. SIRT6 knockdown reduced the HDM/LPS-induced production of N-Ca and increased E-Ca expression, but recombinant human IL-17A (rhIL-17A) reversed these phenomena (Supplementary Fig. 7a, b). Lung fibroblasts are the main source of airway fibrosis that lead to airway remodeling in severe asthma[22]. Thus, we sought to determine the potential role of IL-17A on lung fibroblast in this context. rhIL-17A significantly increased the ECM deposition (eg, α-SMA and collagen I) in fibroblast cells (Supplementary Fig. 7c). To study whether airway epithelial-derived IL-17A promotes ECM deposition, we collected the culture supernatants derived from the HDM/LPS-treated HBE cells. Then we cultured lung fibroblasts with the culture supernatants. We found that treatment with the culture supernatants promoted ECM deposition of lung fibroblasts (Supplementary Fig. 7d-f). Interestingly, the levels of IL-17A were significantly increased the culture supernatants, and neutralizing IL-17A in HDM/LPS-stimulated culture supernatants significantly attenuated ECM deposition, but overexpression SIRT6 almost completely reversed this phenomena (Supplementary Fig. 7g, h).

To further investigate whether blocking IL-17A can prevent airway remodeling in this model, a neutralizing monoclonal antibody against mouse IL-17A (mIL-17A Ab) and/or SIRT6 activator MDL-800 were administered to mice as described in Methods. As predicted, IL-17A expression was significantly higher in HDM/LPS + MDL-800 group compared with the control and HDM/LPS groups, but was significantly lower in HDM/LPS + MDL-800 + IL-17A Ab group. Compared with the control treatment, the blockade of IL-17A also reduced the airway remodeling relative markers such as N-Ca, α-SMA, and collagen I (Supplementary Fig. 7i-l). Collectively, these results suggested that SIRT6-promoted airway remodeling was at least partially mediated by IL-17A.

### SIRT6 directly interactes with RORγt

To investigate the mechanisms by which SIRT6 regulates IL-17A during airway remodeling progress, co-immunoprecipitation (Co-IP) and mass spectrometry (MS) analysis were performed. Using this proteomic approach, 168 potential SIRT6-binding partners were finally

identified (Fig. 5a, Supplementary Fig. 8a, and Supplementary Data 2). We focused on the immune disease-related protein candidates in the KEGG pathway analysis and finally identified five proteins (Fig. 5a). Among them, transcription factor retinoid-related orphan nuclear receptor gamma t (RORγt) is required for initiation of the Th17 program and induction of IL-17A expression[21]. We have therefore completed a further series of experiments to confirm this result. Coimmunostaining validated SIRT6 and RORγt were expressed and colocalization in the HBE cells (Supplementary Fig. 8b). To determine whether SIRT6 and RORγt physically interact, we established co-immunoprecipitation (Co-IP) assays in HDM/LPS-treated HBE cells. The data showed that SIRT6 and RORγt coimmunoprecipitated with each other and that HDM/LPS could up-regulate SIRT6 and RORγt complex (Fig. 5b, c). We further determined whether SIRT6 was able to interact with RORγt exogenously. The exogenous SIRT6-RORγt complex was observed in HBE cells even without HDM/LPS exposure (Fig. 5d). Moreover, the direct interaction of SIRT6 and RORγt was verified by glutathione S-transferase (GST) pull-down (Fig. 5e). Proximity ligation assay (PLA) showed that SIRT6 colocalized with RORγt in the nucleus of HBE cells after HDM/LPS exposure (Fig. 5f). The docked sites of the binding interaction of SIRT6 with RORγt are shown in Supplementary Fig. 8c.

To map the region of RORγt required for its interaction with SIRT6, the proline-rich PPXY motifs of RORγt were constructed as described previous study[21]. It was observed that a highly conserved PPXY motif in RORγt, though which it could potentially interact with SIRT6 (Fig. 5g). We constructed a deletion mutant of amino acids 476-479 of the PPXY motif in RORγt (RORγt-ΔPY) and found that RORγt-ΔPY completely disrupted the SIRT6-RORγt interaction (Fig. 5h). To further determine which of the SIRT6 domain is essential for its interaction with RORγt, three SIRT6 truncations, SIRT6-Core (43-276aa), SIRT6-ΔN (43-355aa), and SIRT6-ΔC (1-276aa) were constructed as described previous study (Fig. 5i, j)[23]. Among the truncations, only the SIRT6-ΔC domain was unable to interact with RORγt (Fig. 5k). Transfection of the SIRT6-ΔC domain, but not SIRT6-wild type (WT), SIRT6-Core, or SIRT6-ΔN domain, was able to decrease RORγt and IL-17A activation (Fig. 5l). Collectively, these results indicated that SIRT6 associated with RORγt via the PPXY motifs and SIRT6-ΔC domain was responsible for its interaction with RORγt in response to HDM/LPS stimulation.

We next assessed whether SIRT6 modulates RORγt expression in the airway epithelium in vivo and in vitro. We first found that RORγt was also largely expressed in airway epithelium cells of asthmatic patients compared with control participants (Fig. 5m). Allergen exposure increased airway epithelial RORγt expression in *AE-Sirt6*[fl/fl] mice, whereas was obviously reduced in *AE-Sirt6*[Δ/Δ] mice compared with *AE-Sirt6*[fl/fl] mice (Fig. 5n). In vitro, allergen exposure induced RORγt expression in HBE cells, and *Sirt6* siRNA reduced RORγt expression compared with the control siRNA, but SIRT6 overexpression almost completely reversed these phenomena (Fig. 5o and Supplementary Fig. 8d, e). In response to allergen stimulation, the levels of RORγt nuclear translocation were significantly reduced in SIRT6-silenced HBE cells compared with control HBE cells (Fig. 5p, q).

### SIRT6 targetes RORγt for deacetylation

Given that SIRT6 is a deacetylation, we investigated whether SIRT6 targets RORγt for deacetylation to inhibit IL-17A expression. We found that HDM/LPS treatment decreased RORγt acetylation in HBE cells, but was inhibited by *Sirt6* knockdown using siRNA approach (Fig. 6a, b). In addition, HEK293 cells were transfected with SIRT6-WT and the three SIRT6 truncations (SIRT6-Core, SIRT6-ΔN, and SIRT6-ΔC). SIRT6 decreased RORγt acetylation, but this phenomenon disappeared after the SIRT6-ΔC domain was deleted (Fig. 6c). Moreover, transfection with the SIRT6-ΔC domain, but not SIRT6-WT, SIRT6-Core or SIRT6-ΔN

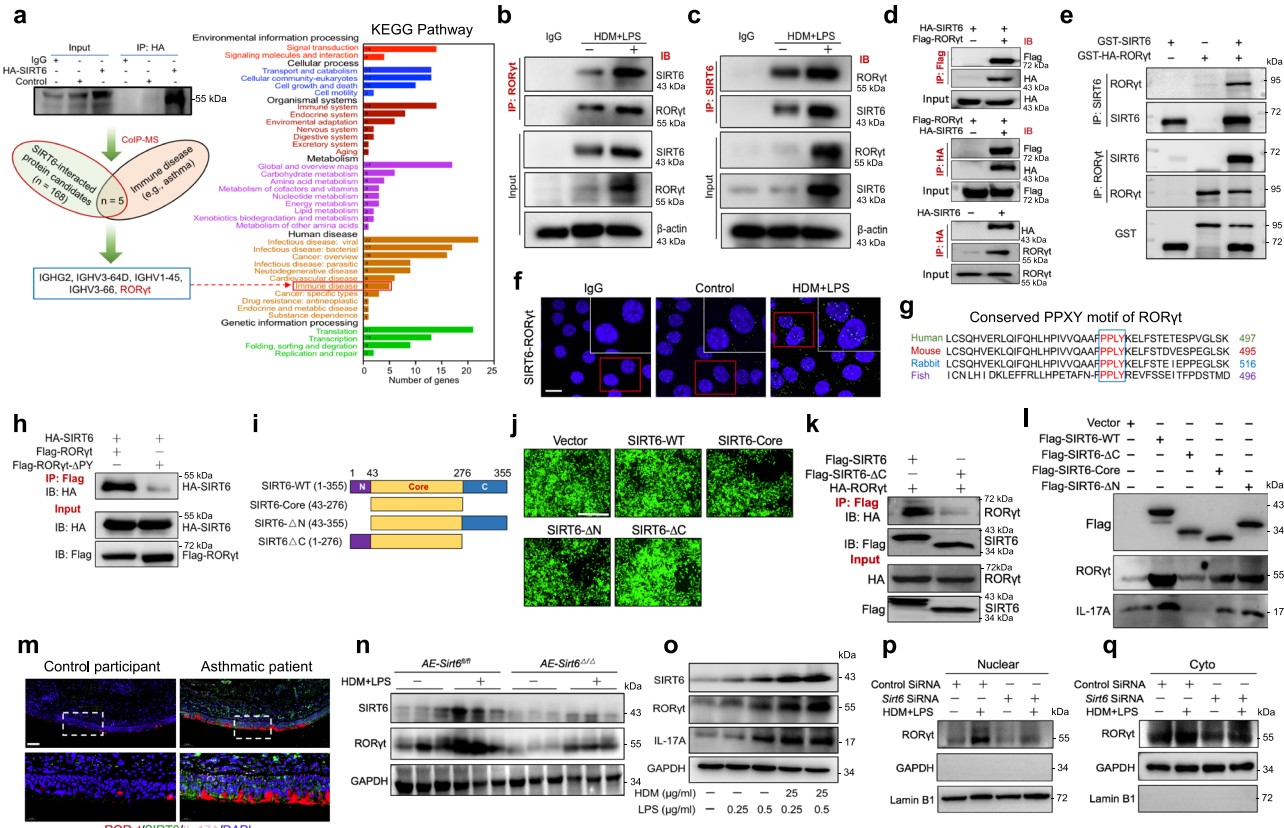

**Fig. 5 | SIRT6 binds directly to RORγt and regulates RORγt expression.**
**a** Experimental flow chart of RORγt discovery by mass spectrometry (MS).
**b**, **c** Endogenous SIRT6 or RORγt in HBE cells was analyzed using Immunoblot (IB).
**d** SIRT6 or RORγt in HBE cells transfected with HA-tagged SIRT6 alone or together with Flag-tagged RORγt was analyzed using Immunoblot (IB). **e** IB analysis of SIRT6 and RORγt interaction in a GST pull-down assay. **f** SIRT6·RORγt molecular interactions in HBE cells was analyzed using Proximity ligation assay (PLA). Foci of interactions are amplified in green (Scale bars, 20 μm). **g** Sequence alignment of RORγt; red indicates the conserved PPXY motif. **h** IB analysis of lysates of HEK293T cells transfected with various combinations (above lanes) of plasmids, followed by immunoprecipitation (IP) with anti-Flag and immunoblot analysis with anti-HA or anti-Flag. **i** Schematic of SIRT6 deletion and point mutants used in this study.
**j** Representative IF staining of GFP-tagged SIRT6 WT and truncations (SIRT6-Core, SIRT6-ΔN, and SIRT6-ΔC) are shown (green) (Scale bars, 400 μm). **k** IB analysis of HEK293T cells transiently transfected with Flag-tagged SIRT6 or Flag-tagged

mutant SRIT6-ΔC domain plus HA-tagged RORγt and assessed 24 h later before (input) or after IP with antibody to Flag. **l** IL-17A expression in HEK293T cells transfected with Flag-tagged SIRT6 WT or Flag-tagged SIRT6 truncations (SIRT6-Core, SIRT6-ΔN, and SIRT6-ΔC) was studied by using Western blot.
**m** Representative confocal IF image of the bronchial tissues from control donor and asthmatic patient stained for RORγt (red), SIRT6 (green), IL-17A (pink), and DAPI (blue) (Scale bars, 100 μm). **n** The expression of RORγt in the lung homogenate of HDM/LPS-exposed *AE-Sirt6^fl/fl* and *AE-Sirt6^Δ/Δ* mice was analyzed by using Western blot. **o** SIRT6, RORγt, and IL-17A expressions in HDM/LPS-treated HBE cells were determined by Western blot. **p**, **q** Western blot analysis of RORγt in cytoplasmic and nuclear extracts of HBE cells transfected with *Sirt6* siRNA for 24 h and then treated with HDM/LPS for another 24 h. GAPDH and Lamin B1 were used as the cytoplasmic and nuclear controls, respectively. Data are representative of three independent experiments with similar results.

domain, was able to decrease IL-17A activation (Fig. 6d, e). To explore the relation of SIRT6 with the IL-17A promoter, we performed an IL-17A promoter-driven luciferase reporter assay. As shown in Fig. 6f, the upregulation of IL-17A promoter activity by SIRT6 overexpression was blocked by the SIRT6-ΔC domain. Together, our results indicated that SIRT6 can promote HDM/LPS-induced RORγt deacetylation through its deacetyltransferase activity. We next performed MS analysis in untreated and HDM/LPS-treated HBE cells to detect acetylated lysine (K) residues of RORγt (Supplementary Data 3). Three acetylated lysine (K) sites (K456, K120, and K192) were identified (Fig. 6g). We then mutated the acetylated lysine (K) sites to arginine (R, mimics deacetylation), and examined their deacetylation (K456R, K120R, and K192R). RORγt acetylation was increased after treatment with nicotinamide (NAM, an inhibitor of the SIRTs). K192R, but not K456R and K120R, resulted in a significant reduction in RORγt acetylation (Fig. 6h). We identified an acetyl-lysine-containing peptide (SYSNN-LAKAGLNGAS) that was mapped to a region containing K192 on human RORγt (Fig. 6i). K192 in RORγt is highly conserved in different species (Fig. 6j), indicating that K192 is the major acetylation site for

RORγt by SIRT6. In addition, SIRT6 targeted RORγt predominantly for K192-linked acetylation (Fig. 6k, l). Acetylation-defective substitution at K192 (K192R) significantly increased HDM/LPS-driven IL-17A protein, *Il17a* mRNA, and *Il17f* mRNA expression, but K192Q (putative acetylation site to glutamine) had no such effect (Fig. 6m, n). Collectively, these results indicated that SIRT6 decreased RORγt acetylation at K192, which played a critical role in the control of IL-17A production.

## SIRT6-RORγt interaction is essential for IL-17A expression
We next performed rescue experiments in SIRT6-KO cell to further elucidate whether SIRT6-RORγt interaction is essential for the promotion of IL-17A expression (Supplementary Fig. 9a, b). Overexpression of the SIRT6-ΔC domain, but not the other two SIRT6 truncations or SIRT6-WT, in SIRT6-KO cells reduced IL-17A expression (Supplementary Fig. 9c). Overexpressing SIRT6-ΔC domain in SIRT6-KO cells, RORγt acetylation could not be detected (Supplementary Fig. 9d). We further investigated whether the up-regulation of RORγt expression in SIRT6-KO cells increased IL-17A expression.

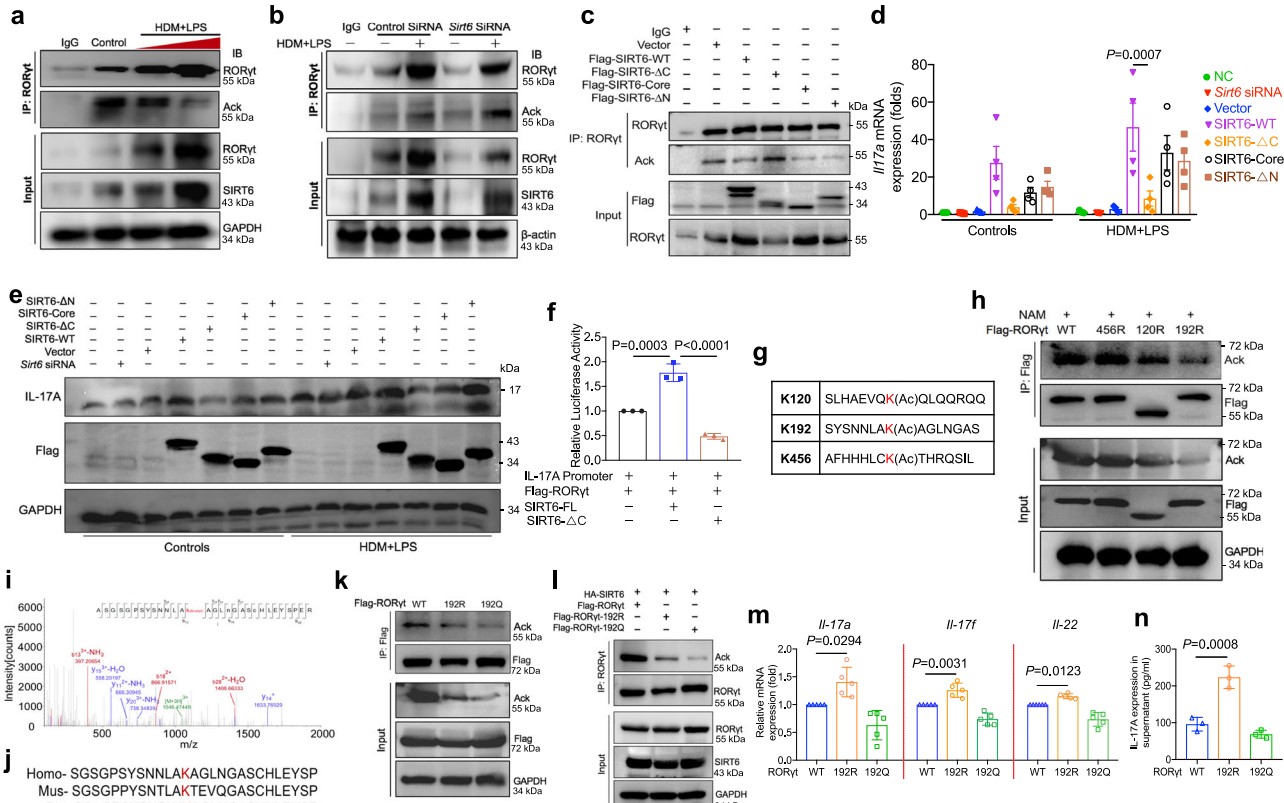

**Fig. 6 | SIRT6 targets RORγt for deacetylation. a** IB analysis of RORγt acetylation (Ack) in HBE cells treated with HDM/LPS and assessed 24 h later before (input) or after IP with antibody to RORγt and Ack. **b** HBE cells were pretreated with *Sirt6* siRNA for 24 h and assessed 24 h later before (input) or after IP with antibody to RORγt and Ack. **c** IB analysis of RORγt Ack in HEK293T cells transfected with WT SIRT6 or SIRT6 truncations (SIRT6-Core, SIRT6-ΔN, and SIRT6-ΔC), assessed before (input) or after IP with antibody to RORγt and Ack. **d, e** qRT-PCR and western blot analysis of *Il17a* expression in HEK293T cells transfected with WT SIRT6 or SIRT6 truncations (SIRT6-Core, SIRT6-ΔN, and SIRT6-ΔC) (n = 4). **f** HBE cells were pre-treated with transfected with WT SIRT6 or SIRT6-ΔC together with RORγt for 24 h and then transfected with the IL17A-reporter plasmid. Luciferase activity was then measured. **g** Identification of acetylated RORγt peptides by mass spectrometry. **h** Analysis of acetylation of individual RORγt mutants. The acetylated lysine (K) sites were mutated to arginine (R, mimics deacetylation) or glutamine (Q, mimics

acetylation). **i** Tandem mass spectrometry of RORγt peptide modified with methylation on lysine 192 residue. **j** Alignment of protein sequences surrounding K192 of RORγt from different organisms. Homo: human, Mus: mouse. **k** Mutation of K192 decreases RORγt acetylation. **l** IB analysis of RORγt Ack in HEK293T cells transfected with HA-tagged SIRT6 WT, Flag-tagged RORγt WT or Flag-tagged RORγt mutant plasmids (K192R and K192Q), assessed before (input) or after IP with antibody to RORγt and Ack. **m** qRT-PCR analysis of *Il17a, Il17f, and Il22* expression in HEK293T cells transfected with K192R and K192Q mutant plasmids (n = 5). **n** The K192R and K192Q mutant plasmids were transfected into HBE cells for 24 h and were then treated with HDM/LPS for 24 h. IL-17A protein were measured using ELISA. Data are shown as means ± SEM and three or more independent experiments were performed. Significance was calculated by one-way ANOVA followed by Tukey's post-hoc test for (**d**−**m**).

We confirmed the RORγt protein expression was increased by Western blot analysis (Supplementary Fig. 9e). Compared to WT cells, RORγt overexpression markedly increased *Il17a* mRNA expression in SIRT6-KO cells, but empty plasmid did not (Supplementary Fig. 9f). RORγt-K192R overexpression in SIRT6-KO cells significantly increased allergen-driven IL-17A expression, but K192Q had no such effect (Supplementary Fig. 9g). To assess the role of RORγt activation on IL-17A expression in SIRT6-KO cells, the RORγt-selective activator LYC-55716 was used. Treatment of SIRT6-KO cells with LYC-55716 increased their reduced secretion of IL-17A (Supplementary Fig. 9h). To confirm that the SIRT6-RORγt interaction is key for IL-17A expression, HBE cells were reconstituted with either RORγt-WT or the RORγt-ΔPY mutant. IL-17A expression was lower in HBE cells reconstituted with RORγt-ΔPY than that of cells reconstituted with RORγt-WT (Supplementary Fig. 9i). To determine the relation of SIRT6 and RORγt with the IL-17A promoter, a chromatin immunoprecipitation (ChIP) assay was performed in the *Sirt6* siRNA- treated HBE cells. HDM/LPS expose promoted RORγt binding to IL-17A promoter. In contrast, this interaction was decreased in SIRT6 siRNA-treated cells (Supplementary Fig. 9j), indicating that SIRT6 downregulation led to reducing RORγt binding to the IL-17A promoter.

## SIRT6 inhibitor treatment attenuates airway inflammation and remodeling

Our data indicated that SIRT6 inhibition might represent a valuable therapeutic approach for severe asthma. We then evaluated the role of OSS_128167 (OSS) (Fig. 7a), a small molecule inhibitor targeting SIRT6[24], in experimental mouse models. As described in Fig. 7b, we first performed an ASA mouse model to assess whether OSS treatment attenuates airway inflammation. H&E staining data demonstrated that OSS treatment attenuated HDM/LPS-induced airway inflammation (Fig. 7c). Total cells and the percentage of eosinophils and neutrophils in BALF of these mice were also decreased (Supplementary Fig. 10a, b). The expression of Ack, SIRT6, IL-17A, and RORγt in the lung homogenates was obviously decreased by OSS administration compared with normal saline administration (Fig. 7d, e). OSS treatment also inhibited inflammatory cytokine production (IL-17A protein, *Il17a* mRNA, *Il17 f* mRNA, *Cxcl1/2* mRNA, and CXCL1/2 protein) in the lung homogenate (Fig. 7f-j). Consistent with animal data, HDM/LPS-stimulated HBE cells showed a significant increase in *Cxcl1, Cxcl2*, and *Il-17a* expression. However, OSS treatment attenuated the secretion of these cytokines (Supplementary Fig. 10c-e).

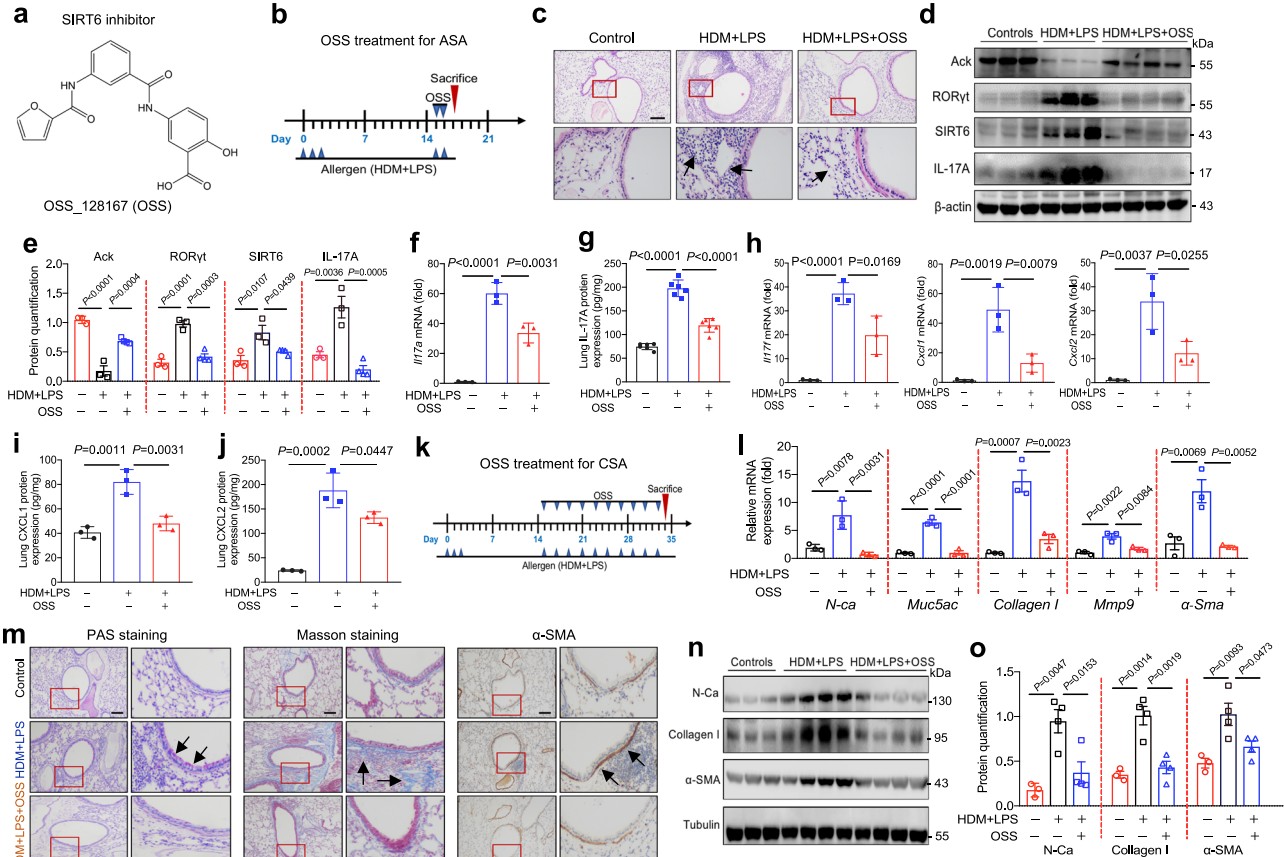

**Fig. 7 | The SIRT6 inhibitor OSS_128167 (OSS) preventes airway remodeling in asthmatic mice. a** Chemical Structure of OSS_128167 (OSS) is shown. **b** Schematic overview of experimental design for (**b–j**) in an ASA model. **c** Representative photomicrographs of lung inflammation expression are shown (Control $n = 4$; HDM + LPS $n = 5$, HDM + LPS + OSS $n = 6$). Scale bars, 100 μm. **d, e** The expression of RORγt, SIRT6, and IL-17A in the lung homogenate from mice treated with HDM/LPS or mice treated with HDM/LPS and SIRT6 inhibitor OSS_128167 (OSS) was analyzed by using Western blot. Quantification was analyzed by Image J software. **f, g** ELISA and qRT-PCR analysis of IL-17A in lung homogenate of mice. **h–j** The expression of inflammatory cytokines in the lung homogenate of mice was analyzed by using qRT-PCR or ELISA analysis. **k** Schematic overview of experimental design for (L-O) in a chronic severe asthma (CSA) model. (Control $n = 4$; HDM + LPS $n = 5$, HDM + LPS + OSS $n = 5$). **l** The expression of *N-ca, Muc5ac, Collagen I, α-Sma,* and *Mmp9* expression in the lung homogenate of mice was analyzed by using qRT-PCR. **m** Representative periodic acid-Schiff (PAS) staining, Masson, and α-SMA of lung sections from WT mice treated with HDM/LPS or HDM/LPS plus SIRT6 inhibitor OSS. Scale bars, 100 μm. **n, o** The expression of N-Ca, Collagen I, and α-SMA in the lung homogenate of mice were analyzed by using Western blot. Quantification was analyzed by Image J software. Data are shown as means ± SEM and three or more independent experiments were performed. Significance was calculated by one-way ANOVA followed by Tukey's post-hoc test for (**e–j–l–o**).

Finally, we asked whether OSS could prevent the progression of airway remodeling. We performed a CSA mouse model as illustrated in Fig. 7k. The production of HDM/LPS-induced *N-ca, Muc5ac, Mmp9, Collagen I,* and *α-Sma* was largely reduced by OSS treatment (Fig. 7l). PAS staining, Masson staining, and IHC demonstrated that OSS treatment attenuated the parameters of airway remodeling (Fig. 7m). These results were confirmed by Western blot (Fig. 7n, o). Thus, our findings suggested that inhibition of SIRT6 may be of potential value in the treatment of airway remodeling and inflammation in severe asthma.

## Discussion

In this study, we revealed that epithelial SIRT6 was essential for IL-17A release, which is an important pro-inflammatory factor in the pathogenesis of allergic airway inflammation and remodeling. SIRT6-ΔC domain directly interacted with RORγt-PPXY motifs and targeted RORγt for deacetylation at lysine 192. The deacetylation RORγt was recruited to IL-17A gene promoter and increased the transcriptional activity. We used a SIRT6 inhibitor, OSS, and found that treatment with OSS protected against allergic airway inflammation and remodeling. Thus, our study disclosed a role for SIRT6 in regulating IL-17A pathogenicity in severe asthma, implicating SIRT6 as a potential therapeutic target for severe asthma.

Epigenetic regulation plays a crucial role in the effects of environmental factors on the pathogenesis of asthma. SIRT6 is an important deacetylase in cells, and can participate in physiological functions such as regulating the expression of various inflammatory genes, DNA damage repair, maintaining telomere stability and aging through the activity of deacetylase[25–33]. However, the function of SIRT6 on disease progression was controversial. Kugel et al. showed that SIRT6 attenuates pancreatic cancer progression and metastasis by regulating Lin28b[13]. Liu et al. indicated that Sirt6 deficiency exacerbates podocyte injury and proteinuria[26]. On the other hand, SIRT6 functions as an oncogene in lung cancer, skin cancer, and multiple myeloma[30–32]. SIRT6 promotes helminth infection-induced tuft and goblet cell differentiation[33]. The airway remodeling is a critical pathologic feature of severe asthma. Airway remodeling process involves a combination of increased EMT, myofibroblast transition, and ECM deposition. Although SIRT6 has been implicated in the development of various diseases, its role in airway inflammation and remodeling in severe asthma is unclear. The airway epithelium is the first line of defense against the external stimulus and secretes many mediators that recruit immune and inflammatory cells to the lung[34]. In our study, SIRT6 was mainly localized to the airway epithelium. Increased epithelial SIRT6 expression was positively correlated with the severity of asthma.

In acute and chronic severe experimental asthma models, we found that both genetic and pharmacological inhibition of SIRT6 attenuated the parameters of airway remodeling, including EMT, myofibroblast transition, and ECM, suggesting that SIRT6 is involved in the pathogensis of severe asthma. Hypoxia-inducible factor 1 alpha (HIF-1α) is the principal regulator of the transcriptional response to allergen[35]. We found that HIF-1α inhibitor BAY87-2243 or HIF-1α knockdown reduced the expression of SIRT6 compared with the control (Supplymentary Fig. S11). Besides TLR4, HIF-1α is also involved in mediating the HDM/LPS-induced SIRT6 expression. However, the potential mechanisms of SIRT6 induction in asthma need further investigation;

Persistent airway inflammation is a crucial risk factor for the airway remodeling process in asthma. IL-17A plays a critical role in EMT process, airway goblet cell hyperplasia, and mucus hypersecretion, which contributes to the development of airway remodeling. A recent research has revealed that mice lacking the IL-17 receptor gene not only exhibit diminished neutrophil recruitment but also reduced eosinophil recruitment into the airways following antigen challenge[36]. Mice deficient in IL-17 or the IL-17 receptor display decreased levels of Th2 cytokines such as IL-5 and IL-13[36,37]. A positive correlation between IL-17 mRNA levels and IL-5 mRNA levels in sputum from asthmatic patients has been established[38]. The inhibition of IL-17 activity using anti-IL-17 antibodies significantly decreases antigen-induced airway infiltration of eosinophils and lowers Th2 cytokine levels in BALF[39]. These results indicated that IL-17A maybe also contributed to Th2 cell-mediated eosinophilic inflammation. In line with these findings, our current data demonstrates that SIRT6 deficiency suppresses IL-17A secretion, resulting in diminished airway infiltration of eosinophils, neutrophils, reduced levels of IL-5, IL-13, and KC/CXCL8 in asthmatic mice. Previous studies have demonstrated that IL-17A can induce the production of MMPs, which contribute to the pathogenesis of airway remodeling[20,21]. The expression of *Mmp3* and *Mmp12* was also decreased in the airway tissues of *AE-Sirt6*$^{\Delta/\Delta}$ mice exposed to HDM/LPS. Taken together, these findings indicated that SIRT6 deficiency inhibited IL-17A expression, which contributing to neutrophilic inflammation and Th2 cell-mediated eosinophilic inflammation. However, Busse et al. showed that there was no evidence of a treatment effect of anti-IL-17 receptor antibody Brodalumab in the full study population of people with asthma[4]. However, it should be noted that Th17-high inflammation (in which high anti-IL17 Brodalumab effects are expected) was not assessed in these patients. Interleukin-17-targeted therapy such as Brodalumab maybe more beneficial in a specific Th17-high asthma patients[40]. Thus, further clinical trials for anti-IL-17 antibody in severe asthma are needed. Morever, we assessed the efficacy of SIRT6-targeted therapy with a inhibitor OSS had effects on the restoration of airway remodeling via inhibiting IL-17A-mediated mesenchymal reprogramming and airway inflammation in acute and chronic experimental asthma. it is also unknown whether OSS is effective in patients with severe asthma and the role of OSS for preventing severe asthma requires further investigation.

Acetylation plays a crucial regulatory role in protein-protein interactions[41]. The transcription factor RORγt plays a critical role in promoting IL-17A transcription[21]. However, the regulatory mechanisms remain unclear. To the best of our current knowledge, no studies have addressed whether and how SIRT6 regulates RORγt expression in severe asthma. The interaction between SIRT6 and RORγt was identified by MS in our study. We discovered that SIRT6 was directly interaction with RORγt and SIRT6-RORγt interaction was essential for IL-17A expression. More and more studies suggest that deacetylation is linked directly to gene transcription. H4K16 deacetylation is required for regulating transcription activation and promotes regional gene expression[42]. Activated SIRT1 deacetylates PGC-1α and subsequently promotes PPARα transcriptional activity, resulting in enhanced HBV replication[43]. In our study, we found that K192 deacetylation of RORγt by SIRT6 induced the recruitment of RORγt to the IL-17A promoter. RORγt deacetylation may influence the interaction between RORγt and cofactors, leading to more abundance of RORγt at the IL-17A promoter. Futhermore, SIRT6 also affected the protein expression level of RORγt and the possible mechanisms are as follows. Protein stability: SIRT6 deacetylation can influence the stability of RORγt. Increased acetylation, as a result of *Sirt6* knockout, may make RORγt more susceptible to proteasomal degradation, resulting in lower protein levels. Acetylation levels: SIRT6 is known to deacetylate RORγt. When Sirt6 is knocked out, the acetylation levels of RORγt may increase, which can lead to changes in its stability and activity. Increased acetylation can promote protein degradation or affect its DNA binding capacity, leading to changes in the targeted protein activity or stability[44,45]. However, the underlying mechanism of SIRT6 affects the protein level of RORγt need to be further studied.

The present,study does have some limitations. First, SIRT6 is expressed in a cell-nonspecific manner (Supplymentary Fig. 2a-h). SIRT6 expression may be upregulated in some cells following exposure to allergen. However, the mechanisms of SIRT6 in different cell types remain further investigation; Second, SIRT6 has been implicated in aging[12]. We found that *AE-Sirt6*$^{\Delta/\Delta}$ mice didn't show aging-associated gray hair, reduced hair density, and hair loss (Supplymentary Fig. 12a). However, the expression of p21 and p16, the number of positive SA-β-gal staining cells were slightly increased in the lung tissues of *AE-Sirt6*$^{\Delta/\Delta}$ mice (Supplementary Fig. 12b, c). Thus, we didn't rule out the dysfunction of epithelial cells due to aging in *AE-Sirt6*$^{\Delta/\Delta}$ and need to be investigated further. Third, although IL-17A expression was increased in the airway epithelial cells in our study, it should be noted that other cell types such as Th17, ILC3 could also be a source of IL-17A. Therefore, the increased expression of IL-17A should be interpreted cautiously due to this limitation.

In conclusion, we identified epithelial SIRT6 to be an important positive regulator of IL-17A secretion in severe asthma, in which it positively regulated airway inflammation and remodeling mainly throug IL-17A-mediated inflammatory chemokines and mesenchymal reprogramming. These data provided insight into the relation of epigenetic regulators with IL-17A production and supplied important clues for a potential strategy against airway inflammation and remodeling in severe asthma.

## Methods

### Ethics statement

Study protocols for research related to human samples including informed consent, publication of demographic data, and associated corresponding source data were approved by the Medical Ethics Committee of Affiliated Hospital of Guangdong Medical University and the Second Hospital Zhejiang University (PJKT2022-079, ChiCTR-OOC-15006345). Written informed consent was obtained from all participants. Experiments were conducted according to the Declaration of Helsinki conventions for the use and care of animals and were approved by the Animal Care and Use Committee at Guangdong Medical University.

### Human study

The diagnosis of asthma and disease severity was based on the Global Initiative on Asthma (GINA) guidelines[1]. Asthmatic patients between 18 and 65 years old were included. Exclusion criteria were combined respiratory diseases other than asthma such as chronic obstructive pulmonary disease (COPD), and lung cancer. Patients who had lung nodules and underwent surgery were used as normal control. Spirometry was done by a turbine spirometry device (Jaeger, Germany) according to the GINA guideline. Asthma control was assessed using the Asthma Control Test (ACT) score. The asthmatic patients were divided into mild, moderate, and severe based on GINA guideline criteria including lung function, ACT score, and drug treatment[1].

## Human specimens

Peripheral blood mononuclear cells (PBMC) were isolated from peripheral blood samples of study participants using density gradient centrifugation according to manufacturer instructions. Bronchial biopsies and BALF were obtained from patients undergoing bronchoscopy for diagnostic purposes. Clinical information is summarized in Table S1 and S2.

## HRCT scanning

HRCT scans of the chest were performed as our previous describe[46]. Single-slice airway measurements were collected in the apical bronchus of the right lower lobe (RB10). By measuring the airway diameter (D) and lumen diameter (L), to calculate the percentage of the total cross-sectional area of the airway wall area (WA%) (WA% = [π (D/2)² - π (L/2)²]/π (D/2)² × 100). WA% was used to compare airway wall thickness between groups (Supplymentary Fig. 13). Measurements were made independently and blind by two experienced radiologists working independently.

## Mice

C57BL/6 mice were purchased from the GemPharmatech Co., Ltd. (Nanjing, China). Sirt6$^{flox/flox}$ (Sirt6$^{fl/fl}$) on the C57BL/6 background were purchased from The Jackson Laboratory. We generated Scgb1a1-rtTA/(tetO)7-Cre/Sirt6$^{fl/fl}$ (designated as AE-Sirt6$^{\Delta/\Delta}$) mice, in which SIRT6 was specifically depleted from airway epithelium cells, by mating Sirt6$^{fl/fl}$ mice with Scgb1a1-rtTA/(tetO)7-Cre transgenic mice (C57BL/6 background). Age-matched and sex-matched Sirt6$^{fl/fl}$ mice were used as controls in the experiments. To induce the expression of Cre, 6-week-old mice were fed with doxycycline (DOX) in drinking water (2 mg/ml) for 20 days before establishing the model of allergen-induced airway inflammation, and the mice were kept with DOX at all the time until they were sacrificed. All animals were age-matched and sex-matched, and then randomized into different groups. The age for all strains of mice are 6–8 weeks. All mice were maintained in specific pathogen-free animal facilities at the Animal Care Facility of Guangdong Medical University. The housing conditions for the mice like dark/light cycle is 12 h, the ambient temperature is 20–25 degrees centigrade and the relative humidity is 40-70%. Experiments were conducted according to the Declaration of Helsinki conventions for the use and care of animals and were approved by the Animal Care and Use Committee at Guangdong Medical University.

## Animal studies

Severe asthma patients have eosinophilic inflammation or neutrophilic inflammation or mixed granulocytic (both eosinophilic and neutrophilic inflammation). Common indoor and outdoor environmental exposures can influence airway inflammation in asthma. The presence of endotoxin during allergen sensitization can lead to neutrophilic airway inflammation in mice. HDM is a complex mixture, containing mite allergens and microbial products. LPS is an endotoxin derived from the membrane of colonizing Gram-negative bacteria and environmental contamination, and the level of LPS is correlated with the severity of asthma and decline in lung function[17]. Co-exposure to HDM and LPS has been used to establish neutrophil-dominated airway inflammation, mucus hypersecretion, and airway remodeling model that effectively simulates the pathological and physiological processes of clinical severe asthma patients[47–50]. We found that HDM or LPS alone activated IL-17A expression in a dose-dependent manner (Supplementary Fig. 10a, b). IL-17A production was slightly increase in HBE cells exposed to low dose LPS (0.125 μg/ml) alone, but HDM significantly potentiated low dose LPS-induced IL-17A expression (Supplementary Fig. 10c). Accordingly, severe asthma mouse model was adapted in this study as previously described[47–49]. For acute severe asthma (ASA) model,

Sirt6$^{fl/fl}$ and AE-Sirt6$^{\Delta/\Delta}$ mice were sensitized with house dust mite (HDM) combined with lipopolysaccharide (LPS) on days 0, 1, and 2 by intratracheal instillation of (5 μg HDM + 3 μg LPS)/50 μl. On days 15, 16 after the initial sensitization, the mice were challenged with (2.5 μg HDM + 1.5 μg LPS)/50 μl using intratracheal instillation. For chronic severe asthma (CSA) model, Sirt6$^{fl/fl}$ and AE-Sirt6$^{\Delta/\Delta}$ mice were sensitized on days 0, 1, and 2 by intratracheal instillation of (5 μg HDM + 3 μg LPS)/50 μl. On days 15, 17, 19, 21, 23, 25, 27, 29, 31, and 33 after the initial sensitization, the mice were challenged with (2.5 μg HDM + 1.5 μg LPS)/50 μl using intratracheal instillation. For SIRT6 inhibitor OSS_128167 (OSS) treatment, OSS, at concentrations of 10 mg/kg, was given via intraperitoneal injection 2 h before challenged with HDM/LPS. For SIRT6 activator (MDL-800) and IL-17A neutralization (IL-17A Ab), WT mice were divided into four groups: (i) Control, (ii) HDM + LPS, (iii) HDM + LPS + MDL-800, and (iv) HDM + LPS + MDL-800 + IL-17A Ab. Mice were sensitized on days 0, 1, and 2 by intratracheal instillation of (5 μg HDM + 3 μg LPS)/50 μl. On days 15, 17, 19, 21, 23, 25, and 27 after the initial sensitization, the mice were challenged with (2.5 μg HDM + 1.5 μg LPS) /50 μl using intratracheal instillation. On days 15, 19, 23, and 27, MDL-800 (5 mg/kg via intratracheal instillation) with and without IL-17A Ab (1.5 μg via intratracheal instillation) was given 2 h before challenged with HDM/LPS.

## Collection of bronchoalveolar lavage fluid (BALF)

BALF was performed as described in our previous study[46]. Briefly, the lungs were administered lavage using a cannula inserted in the trachea, and then they were instilled with 0.8 ml of phosphate-buffered saline (PBS) after exsanguination. This process was repeated three times to collect about 2 mL of BALF from each mouse. Cytospin slides were prepared by Wright-Giemsa staining. Total cell and differential cell count in BALF were counted under the microscope in a blinded method.

## Histological analysis and immunohistochemistry (IHC)

Lung sections were stained with hematoxylin/eosin (HE) for airway inflammation, Periodic Acid-Schiff (PAS) for mucus production, and Masson trichrome for collagen deposition. The slides were incubated with antibodies against SIRT6, IL-17A, RORγt, or α-SMA. Sections were examined using a light microscope (BX43; Olympus, Japan) and quantified by the Image J software.

## Immunofluorescence (IF)

The cell slides were fixed with 4% paraformaldehyde for 25 min and then permeated with 0.1% TritonX-100 at room temperature for 10 min. After washing with PBS and sealing with 5%BSA. The nucleus was stained with DAPI diluent. All of the section samples used for the analysis were viewed using confocal microscopy (Olympus FV3000; Japan) at ×400 magnification and digitalization software. The percentages of SIRT6&SCGB1A1/DAPI, IL-17A&SCGB1A1/DAPI co-labeled cells in the total airway epithelium cell (SCGB1A1) was quantified by Image J software. An individual blinded to experimental conditions did all the counting.

## Cell culture

The HBE cells used in this experiment were obtained from the Center for Typical Culture Preservation in the United States and introduced into the Institute of Respiratory Diseases, Zhejiang University. Primary fibroblasts used in this study were isolated and cultured from the lung tissues of C57BL/6 mice. HBE cells, HEK293T cells and primary mouse fibroblasts were cultured with DMEM containing 10% ～ 20% FBS (Gibco).

## Isolation of primary fibroblasts

Primary fibroblasts were isolated described previously by our laboratory[44]. Cells were obtained from lung tissues of WT mice.

Identification of fibroblasts was based on the expression of vimentin. All experiments were carried out using cells between passages 3 and 6.

## Plasmid constructs and transfection

Mutants of Sirt6 were constructed according to the previously described[23]. Briefly, Sirt6 (1-276aa), Sirt6 (43-276aa), Sirt6 (43-334aa), and Sirt6 (1-355aa) were generated by PCR cloning into GV362 eukaryotic expression vector and digested by XhoI/BamHI enzyme purchased from Shanghai Genechem Co., LTD. The plasmid HA-SIRT6 was created from Sirt6 (1-355aa) and cloned into the pcDNA3.1-HA vector. pGEX-4T-2-HA-RORγt, pCAGPuro-RORγt-HA, RORγt-D-PPLY, RORγt-K192Q, RORγt-K192R, and GST-SIRT6 plasmids were generated by using the templates RORγt (Shanghai Yubo Biotechnology Co., LTD; China). was also purchased from Shanghai Yubo Biotechnology Co., LTD. The *Sirt6* siRNA used in the experiment was synthesized by Genepharma, and the sequence (5′-3′) was as follows: UCCAUC ACGCUGGGUACA UTT; AUGUACCCAGCGUGAUGGATT. For transient transfection of plasmids and *Sirt6* siRNA in HEK293T cells or HBE cells, GP-transfection-Mate (220308; Genepharma) was used according to the manufacturer's instructions.

## Quantitative real-time PCR

Total RNA was extracted from cells using TRIzol reagent (Invitrogen) following the instructions. The analysis was performed using Light Cycler 480 (Roche) and a SYBR RT-PCR kit (Takara). The primer sequences were shown in the Table S3.

## Luciferase assay

Firefly luciferase substrate (LARII), passive lysis buffer (1 × PLB) and STOP GLO reagent were prepared according to the kit instructions. 48 h after transfection, discard the medium, wash twice with PBS buffer, add 1 × PLB 500 μl/well to 6-well plate, and lysis at room temperature for 20 min on shaking table, then suck the mixture into 1.5 ml EP tube. Centrifugation at 4 °C, 12000 × g, 5 min, remove the supernatant and transfer it to a new 1.5 ml EP tube. Take 10 μl supernatant for detection, add 50 μl LARII working solution first, mix and quickly put into the instrument for detection, record firefly luciferase activity F value, then add 50 μl STOP GLO reagent, mix and detect, record sea renal luciferase activity R value and calculate the ratio F/R value.

## ChIP assay

ChIP detection kit (Beyotime, China) was used for ChIP detection, and the experiment was carried out according to the instructions. In a nutshell, cells are treated with 1% formaldehyde to cross-link DNA and proteins. The cell lysates were then ultrasonically lysed for 10 times to generate 200 bp-300 bp chromatin fragments, and immunoprecipitation was performed with anti-RORγt antibodies. The precipitated chromatin DNA was purified and recovered, and the products amplified by qPCR were subjected to agarose gel electrophoresis.

## GST pull-down

GST-SIRT6 and GST-HA-RORγt plasmids were constructed. The GST fusion protein was expressed in EScherichia coli BL21 cells and purified using GST-Tag protein purification Kit (Beyotime) according to the manufacturer's instructions. In the experiment, GST-SIRT6 and GST-HA-RORγt proteins were mixed and then the corresponding primary antibody was added and incubated overnight at 4 °C. On the 2nd day, each sample was added with the same amount of Protein G Beads, and incubated at 4 °C for 1 - 2 h. Then, the samples were then washed and immunoblotted using anti-SIRT6 or anti-RORγt antibody.

## Western blotting

The samples were separated by New Semet Express-cast PAGE color gel electrophoresis and electrically transferred to PVDF membranes. The membranes were incubated with primary antibodies overnight at 4 °C after blocking with 5% fat-free milk for 1 h. The PVDF membrane was washed with 1 × TBST solution for 3 times, and then incubated with the corresponding secondary antibody for 1 h. Then the PVDF membrane was washed with 1 ×T BST solution 3 times again, and the exposure was analyzed and sorted out.

## Co-immunoprecipitation (Co-IP)

After Total proteins of cells were extracted with cell lysis buffer, the protein concentration was determined by the BCA method and 1 mg protein was taken for subsequent immunoprecipitation. Each sample was incubated with 1 μg antibody and 30 μl Protein G Beads suspension successively, centrifuged at low temperature and washed, then resuspended with 2 × SDS Sample Buffer for Western blot.

## ELISA

Human IL-17A, mouse IL-17A, CXCL1, CXCL2, IL-5, and IL-13 were measured using ELISA kit (Elabscience, China). Assays were according to the manufacturer's protocol.

## CRISPR/Cas9 KO cells construction

The plasmid required for SIRT6-KO cell construction was obtained from Santa Cruz. HEK293 cells were placed on 12-well plates. On the second day, 10 μl SIRT6 CRISPR/Cas9 KO plasmid at 0.1 μg/μl was transfected into HEK293 cells. The expression of green fluorescence was observed 48 h later. After that, the green fluorescence expression of the cells was observed every day, and the fluorescence was essentially stable for about 20 days after subculture. All transfected cells were gathered for flow sorting after stable fluorescence expression, and cells with strong green fluorescence were sorted out for enrichment and culture. After the cells were cultured and subculture, the knockout efficiency of SIRT6 was detected by WB and qRT-PCR.

## Flow cytometry

Lung tissues of mice were prepared into cell suspension, and appropriate amount of surface marker antibodies were added according to the instructions, and incubated for 30 min on ice, protected from light. Flow cytometry data were obtained using Fortessa flow cytometry (BD FACSCantoII) and analyzed using FlowJo software (Becton Dickinson). Single-cell suspensions were stained with combinations of the following antibodies (Biolegend): anti-CD3 (APC/Cyamine7), anti-CD4 (FITC), anti-F4/80 (APC), anti-CD49b (PE), anti-CD11c (PE), and anti-CD19 (Percp/Cyamine7).

## Mass spectrometry (MS)

The immunoprecipitate was digested overnight at 37 °C with trypsin and then lyophilized to dryness. The peptide samples were analyzed on Thermo Fisher LTQ Obitrap ETD mass spectrometry. Mass spectrometry analysis were carried out carried out at the AIMSMASS Co.,Ltd. (Shanghai, China). The MS/MS was acquired using higher-energy collision dissociation at 35% collision energy at a mass resolution of 15,000. Raw MS files were analyzed by MaxQuant (version 1.5.2.8), the parameter used for data analysis included trypsin as the protease with a maximum of two missed cleavages allowed.

## RNA-Seq analysis

Total RNA was extracted using Trizol reagent kit (Invitrogen, Carlsbad, CA,USA) according to the manufacturer's protocol. RNA quality was assessed on an Agilent 2100 Bioanalyzer (Agilent Technologies, Palo Alto, CA, USA) and checked using RNase free agarose gel electrophoresis. The resulting cDNA library was sequenced using Illumina Novaseq6000 by Gene Denovo Biotechnology Co. (Guangzhou, China).

## In situ proximity ligation assay (PLA)

HBE cells permeabilized with 0.1% Triton X-100, and incubated with IgG or primary MS rabbit antibody to SIRT6and mouse antibody to RORγt.

Slides were then subjected to PLA by using the Duolink Detection Kit (Cambridge BioScience Ltd), according to the manufacturer's instructions. Nuclei were counterstained with DAPI, and PLA signals were visualized in a LSM510 fluorescence confocal microscopetio.

### Senescence-associated β-galactosidase staining

Paraffin sections were mounted onto glass slides and Deparaffinizing and rehydrating the section. Then fixed in 0.2% glutaraldehyde and 2% formaldehyde at room temperature for 15 min. Sections were washedin PBS and incubated in freshly prepared senescenceassociated β-galactosidase (SA-β-gal) staining solution (1 mg/mL X-gal, 40 mM citric acid/sodium phosphate (pH 6.0), 5 mM potassium ferrocyanide, 5 mM potassium ferricyanide, 150 mM NaCl, and 2 mM $MgCl_2$) overnight at 37 °C without $CO_2$. Tissue sections were counterstained with eosin and examined under a microscope.

### Statistical analysis

All statistical analyses and figures were performed using GraphPad Prism 8.0 (San Diego, CA). Unless stated otherwise, data are presented as means ± SEM. Variable differences between two experimental groups were assessed using Two-tailed unpaired Student's $t$-test. An ordinary one-way analysis of variance (ANOVA) followed by Tukey's post hoc test for the comparison of multiple groups. Categorical variables was tested using Chi-square. A $P$-value of <0.05 was considered statistically significant.

### Reporting summary

Further information on research design is available in the Nature Portfolio Reporting Summary linked to this article.

## Data availability

The data that support the findings of this study are available within the article and its supplementary information or from the corresponding authors on request. Key resources are shown in Table S4. RNA-seq data that support the findings of this study have been deposited in Genome Sequence Archive with the accession number SRP454168. Source data are provided with this paper.

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

## Acknowledgements

We would like to thank Zhihua Chen, Chao Zhang, and Xufei Du from Zhejiang University; Chen Yang, Lianxiang Luo, Xingdong Xiong, and Lili Cui from Guangdon Medical University for comments on the manuscript and helpful discussion. We would like to thank Guihai Pan from Guangdon Medical University for analysis of the HRCT images. This research was supported by the Guangdong Basic and Applied Basic Research Foundation (2020B1515020004), National Natural Science Foundation of China (82170030; 81873404), Guangdong Provincial Key Laboratory of Autophagy and Major Chronic Non-communicable Diseases (2022B1212030003).

## Author contributions

J.Y.Q., X.X.W., G.M.S., Y.Z., and T.H. conducted experiments, data analysis. Z.L.X., J.W.H., Y.Y.L., S.H.L., S.H.L, X.W.L., and Y.Y.X. performed data analysis. C.L.L. and X.C. assisted with the experiments and clinical samples collection, data analysis. T.W.L., J.T., H.T.L., S.G.Z., Y.M.S., and X.G. analyzed/interpreted results and edited the manuscript. T.W.L. conceived, designed, and supervised the whole study. T.W.L. and J.Y.Q. wrote the manuscript. All authors read and approved the final manuscript.

## Competing interests

The authors declare no competing interests.
