## [Peer Review File · Nature Communications]

Epithelial SIRT6 governs IL-17A pathogenicity and drives allergic airway inflammation and remodelingREVIEWER COMMENTS

Reviewer #1 (Remarks to the Author):

Thank you for giving me the opportunity to review this paper. The authors have shown that Sirt6 induces IL-17 production via ROR γ t and is involved in the severity of asthma. The amount of experiments is extensive and the molecular mechanism is well investigated. However, several critical questions remain when considering its clinical application. Sirt6 is associated with aging, and progeria has been reported to occur in Sirt6 knockout mice. Is it really beneficial for the patient to suppress Sirt6 in order to suppress IL-17? While the discovered mechanism is interesting, it cannot be accepted unless the following questions are resolved.

Major points

1. Discrepancies between animal models and asthma patients

In this paper, HDM and LPS were administered as a model of acute severe asthma. The pathogenesis of severe asthma is very diverse, and it is questionable whether this model is appropriate as a model for severe asthma. Anti-IL-17 receptor antibody was not effective against severe asthma (AJRCCM, 2013), so I doubt that this pathway will be therapeutic.

2. Mechanism of Sirt6 induction

Does administration of HDM or LPS alone induce Sirt6? It should be clarified what stimuli induce Sirt6 and by what mechanism. Could it be just TLR4-dependent? If so, would that change the story?

3. Differences in cell types

It is inconsistent because Sirt6 is expressed in airway epithelial cells in mice, while in peripheral blood in human (Figure 1). Is Sirt6 elevated in a cell-nonspecific manner? Different cells might be experiencing different phenomena?

4. Sirt6-expressing cells

Sirt6-expressing cells should be evaluated more comprehensively. For example, what about the expression of alveolar epithelial cells in mice? If they are expressed, it should be shown

how expression is altered in your conditional knockout mice. Hematopoietic cells only looked at macrophages. Which cells have elevated Sirt6 in humans? The authors may be missing the cells that are really expressing Sirt6.

5. Conditional knockout mice

In your mouse model, eosinophils are decreased in the conditional knockout mice. I don't think it is an effect of IL-17. Could Sirt6 have other important functions besides IL-17 induction? In particular, Sirt6 has been implicated in aging, and Sirt6 knockout mice show progeria and die prematurely. Is there a dysfunction of epithelial cells due to aging in these mice?

6. OSS

Does OSS have an inhibitory effect in human cells? Is it really beneficial to suppress Sirt6 with OSS for patients with severe asthma since even IL-17 antibodies were ineffective?

Minor points

1. Allergen-induced HBE cells sound strange.
2. Figure 7E should be labeled.

Reviewer #2 (Remarks to the Author):

The authors have shown a novel role of SIRT6 as an epigenetic regulator of IL-17A production in bronchial epithelial cells via deacetylation of ROR γ t-K192. The strength of this study is that they demonstrate that SIRT6 inhibition in bronchial epithelium reduces airway inflammation and prevents airway re-modelling in an acute severe asthma mouse model. In addition, most findings were linked to asthma patients comparing to controls. However, there are some major and minor comments to be addressed as follows.

Major comments:

1. In figure 1F-G, show that Sirt6 mRNA is increased peripheral blood of asthma patients and even higher in severe asthma patients. The main findings of this manuscript highlights that Sirt6 specifically from bronchial epithelium increases IL-17A, so the authors could

investigate the expression of Sirt6 in bronchial brushes comparing mild-moderate asthma patients and severe asthma patients with non-asthmatic controls. Therefore comparison of systemic increase of Sirt6 in peripheral blood may not translate to increased Sirt6 expression in bronchial epithelial cells.

2. In figure 2O, Sirt6+ cells in airway epithelium was compared in airway biopsies between asthma patients and controls. Here, the Sirt6+ cells were comparable between severe asthma patients and mild-moderate asthma patients, even though the numbers of donors are not sufficient for this conclusion. The authors should elaborate the method of measuring Sirt6+ cells in airway epithelium in the figure legends or in the methods section. The y-axis shows 'Sirt6+ cells in airway epithelium (% of total cells)', if the total number of cells in the tissue section is different (as seen in Figure 2N), it may impact the analysis. The method should also include how the airway epithelium was distinguished from other cell types, for instance, airway smooth muscle cells. This also applies to Figure 3K, where IL-17A+ cells in airway epithelium is shown.

3. In Figure 1I, the authors correlate FEV1 to Sirt6 mRNA expression in all donors including controls, mild-moderate and severe asthma patients. To correlate the Sirt6 mRNA expression the correlation must be done only using mild-moderate and severe asthma patients and exclude controls. An example of this analysis is in Figure 1K, showing correlation of ACQ scores with Sirt6 mRNA expression.

4. The characteristics of asthma patients in the peripheral blood study has been shown in Table S1. The percentage of eosinophils are higher in severe asthma patients compared to mild-moderate and controls. IL-17A levels in severe asthma patients is linked with neutrophilic inflammation. Therefore, it is relevant to show, if percentage of neutrophils are higher in severe asthma patients compared to the other 2 groups. In addition, depicting correlation between Sirt6 mRNA levels and neutrophils percentage, would strengthen the link between IL-17A and neutrophilic inflammation.

5. In figure 3K, the authors demonstrate decrease of eosinophils, neutrophils and lymphocytes in BALF of AE-Sirt6 Δ/Δ mice treated with HDM and LPS. As this an allergic

mouse model did the levels of IL-5, IL-13 (eosinophilic inflammation) and CXCL-8 (neutrophilic inflammation) also decrease in the BALF? Nonetheless, it is impressive that the knockout of Sirt6 specifically in bronchial epithelium, leads to overall decrease in airway inflammation and remodelling.

6. It has been previously shown, that the major source IL-17A are the Th17 cells. Therefore, is IL-17A+ CD4+ T cells increased in asthmatic mice compared to control mice (in Sirt6fl/fl mice)?

In Figure 4E, IL-17A in the BALF is increased, the source of this IL-17A could also be from Th17 cells. Considering in Figure 4B, Th17 immune response is increased in asthmatic mice (as shown by RNA-seq of airway tissue), Th17 could also be a source of IL-17A, apart from bronchial epithelium.

There are also, other cell types that produce IL-17A, for instance ILC3, can the authors comment on this.

7. In Figure 4Q, HDM in combination with LPS increase IL-17A production in in vitro culture HBE. Does HDM or LPS alone have the capacity to increase IL-17A production? Is the increased IL-17A produced by HBE a synergistic effect i.e. observed only in combination or does LPS or HDM alone increase IL-17A?

LPS alone is a potent stimuli, so is HDM required to see this increased IL-17A production? If not, then the in vitro effects observed in HBE does not pertain to HDM induced allergic airway inflammation.

8. In Figure 7C, Sirt6 inhibitor reduced airway inflammation as shown by H&E staining. In BALF of these mice, were the percentage of eosinophils and neutrophils also decreased?

9. Severe asthma patients have eosinophilic inflammation or neutrophilic inflammation or mixed granulocytic (both eosinophilic and neutrophilic inflammation). The authors can add to the discussion on the effectiveness of the Sirt6 inhibitor as a therapeutic for severe asthma with predominant eosinophilic inflammation. Does inhibiting IL-17A be effective in reducing eosinophilic inflammation in severe asthma patients?

Minor comments

1. In figure 1C, Sirt1, 3, 5, expression is decreased in HDM sensitized and challenged mice. Is this decrease also observed in asthma patients compared to controls in peripheral blood?

Reviewer #3 (Remarks to the Author):

This is a very interest finding about regulation of IL-17A in severe asthma. results showed the role of SIRT6 in regulating IL-17A pathogenicity. But still some problems should be concerned.

1. In Figure1 animal model section, whether there is validation on the success of the animal modeling, such as HE, IHC, type II inflammatory factor determination, etc.
2. In Figure1, in the part of human experiment, whether there is protein level verification of PBMC to determine the change of sirt6, and whether there is mRNA and protein verification of human lung tissue.
- 3, Whether the enzymatic activity of sirt6 has changed in the mouse models and asthma patient.
4. In Figure2 A, the expression of sirt6 seems to be inconsistent in the same lung tissue, is there any bias in the selection of view?
5. In lines 160 and 162, the figure labels do not match the picture labels.
6. In Figure5D, why not set a group which is only transfected with HA-SIRT6, not transfected with Flag-RORrt?
7. In Figure5K, the co-IP results show that the SIRT6- Δ C group still binds to RORyt, which is not consistent with the conclusion.
8. Please label the antibodies used in the text, such as the brand and product number of the AcK antibody used for acetylation detection in figure 6.
9. In Figure6D, different sirt6 truncations affect the expression of IL17, whether there is WB protein level result?
10. In Figure6G, 6J, no input results are provided, such as IB Flag, Ace- α -tubulin, internal reference protein tubulin. In Figure6K, IB RORyt is missing in IP, and the result of internal reference protein is missing in input.

The point-by-point responses to the comments

Reviewer: 1

Major points

1. Discrepancies between animal models and asthma patients

In this paper, HDM and LPS were administered as a model of acute severe asthma. The pathogenesis of severe asthma is very diverse, and it is questionable whether this model is appropriate as a model for severe asthma. Anti-IL-17 receptor antibody was not effective against severe asthma (AJRCCM, 2013), so I doubt that this pathway will be therapeutic.

R: Thank you very much for your sincere advices. Severe asthma patients have eosinophilic inflammation or neutrophilic inflammation or mixed granulocytic (both eosinophilic and neutrophilic inflammation). Common indoor and outdoor environmental exposures can influence airway inflammation in asthma. The presence of endotoxin during allergen sensitization can lead to neutrophilic airway inflammation in mice. HDM is a complex mixture, containing mite allergens and microbial products. LPS is an endotoxin derived from the membrane of colonizing Gram-negative bacteria and environmental contamination, and the level of LPS is correlated with the severity of asthma and decline in lung function^[1]. Co-exposure to HDM and LPS has been used to establish neutrophil-dominated airway inflammation, mucus hypersecretion, and airway remodeling model that effectively simulates the pathological and physiological processes of clinical severe asthma patients^[2-6]. Accordingly, a mouse model of severe asthma was developed by HDM and LPS in our study. We found that the total inflammatory cells, neutrophils, eosinophils, and lymphocytes in bronchoalveolar lavage fluid (BALF) were reduced in HDM/LPS-exposed *AE-Sirt6*^{Δ/Δ} mice (Fig. A and B). HE staining and quantitative analysis showed that SIRT6 deficiency attenuated HDM/LPS-induced peribronchial infiltrates of inflammatory cells (Fig. C and D). Several features of airway remodeling examined by peribronchial trichrome (Masson) staining, periodic acid Schiff (PAS) staining, and α -smooth muscle actin (α -SMA) were reduced in HDM/LPS-exposed *AE-Sirt6*^{Δ/Δ} mice (Fig. E and F). The levels of

eosinophilic inflammation (IL-5, IL-13) and neutrophilic inflammation (KC/CXCL-8) were also reduced in HDM/LPS-exposed *AE-Sirt6*^{Δ/Δ} mice (Fig. G-I). Taken together, these findings indicated that this model is appropriate as a model for severe asthma.

Busse et al. showed that there was no evidence of a treatment effect of anti-IL-17 receptor antibody Brodalumab in the full study population of subjects with asthma^[7]. Because asthma is a heterogeneous disease with clinical subphenotypes that respond to specific biologic therapy within a larger population, these findings are not necessarily unexpected. However, in a subgroup of patients with high FEV₁ reversibility to inhaled bronchodilator ($\geq 20\%$ improvement; $n = 112$). It should be noted that Th17-high inflammation (in which high anti-IL17 Brodalumab effects are expected) was not assessed in these HDM+LPS patients. Moreover, the study populations were highly atopic (83% in total population; 79 and 84% in placebo and Brodalumab treatment arms, respectively). It has been reported that atopic asthma is more likely to correlate to an eosinophilic phenotype, therefore the subjects included in the study might not be the best target group for Brodalumab^[7]. Interleukin-17-targeted therapy such as Brodalumab maybe more beneficial in a specific Th17-high asthma subjects. It is unknown whether Brodalumab would have been more efficacious if patients had been selected for study using a phenotype-specific approach, that is, the presence of significant sputum neutrophilia or absence of Th2 biomarkers^[8]. Thus, further clinical trials for anti-IL-17

antibody in severe asthma are needed and should consider these influencing factors and requires further investigation. We have discussed it in the revised version.

We don't know whether we answered your question and sincerely hope you can give us some suggestions. We really appreciate your earnest and rigorous work attitude and it is worthy of our learning.

References

1. Daan de Boer J, Roelofs JJ, de Vos AF, et al. Lipopolysaccharide inhibits Th2 lung inflammation induced by house dust mite allergens in mice. *Am J Respir Cell Mol Biol.* 2013;48:382-9.
2. Krishnamoorthy N, Douda DN, Brüggemann TR, et al. Neutrophil cytoplasts induce TH17 differentiation and skew inflammation toward neutrophilia in severe asthma. *Sci Immunol.* 2018;3:eaa0 4747.
3. Wang L, Netto KG, Zhou L, et al. Single-cell transcriptomic analysis reveals the immune landscape of lung in steroid-resistant asthma exacerbation. *Proc Natl Acad Sci U S A.* 2021 Jan 12;118:e2005590118.
4. Daan de Boer J, Roelofs JJ, de Vos AF, et al. Lipopolysaccharide inhibits Th2 lung inflammation induced by house dust mite allergens in mice. *Am J Respir Cell Mol Biol.* 2013;48:382-9.
5. Michel O, Kips J, Duchateau J, et al. Severity of asthma is related to endotoxin in house dust. *Am J Respir Crit Care Med.* 1996;154:1641-6.
6. Eisenbarth SC, Piggott DA, Huleatt JW, et al. Lipopolysaccharide-enhanced, toll-like receptor 4-dependent T helper cell type 2 responses to inhaled antigen. *J Exp Med.* 2002;196:1645-51.
7. Busse WW, Holgate S, Kerwin E, et al. Randomized, double-blind, placebo-controlled study of brodalumab, a human anti-IL-17 receptor monoclonal antibody, in moderate to severe asthma. *Am J Respir Crit Care Med.* 2013;188:1294-302.
8. Corren J. New Targeted Therapies for Uncontrolled Asthma. *J Allergy Clin Immunol Pract.* 2019;7:1394-1403.

2. Mechanism of Sirt6 induction

Does administration of HDM or LPS alone induce Sirt6? It should be clarified what stimuli induce Sirt6 and by what mechanism. Could it be just TLR4-dependent? If so, would that change the story?

R: We appreciate the constructive criticism and suggestions. As your suggestion, we detected SIRT6, IL-17A, and TLR4 expression in human bronchial epithelial (HBE) cells after HDM or LPS stimulation. We found that HDM or LPS activated SIRT6, IL-17A, and TLR4 expression in a dose-dependent manner (Fig. A and B). HDM significantly potentiated LPS-induced SIRT6, IL-17A, and TLR4 expression (Fig. C).

Recent studies have implicated that Hypoxia-inducible factor 1 alpha (HIF-1 α) plays an integral role in the pathogenesis of asthma. HIF-1 α is the principal regulator of the transcriptional response to allergen^[1,2]. Therefore, we hypothesized that HIF-1 α might mediate SIRT6 expression exposed to HDM/LPS. To test this, we analyzed the effect of HDM/LPS on HIF-1 α expression in HBE cells. We found that HIF-1 α protein was significantly increased in HBE cells exposed to HDM/LPS (Fig. C). However, HIF-1 α inhibitor BAY87-2243 or HIF-1 α knockdown reduced the expression of SIRT6 compared with the control (Fig. D-F).

Collectively, we found that administration of HDM or LPS alone can induce the expression of SIRT6, IL-17A, and TLR4 in HBE cells. Besides TLR4, HIF-1 α is also involved in mediating the HDM/LPS-induced SIRT6 expression. However, the potential mechanisms of SIRT6 induction in asthma need further investigation. We have added this result and discussed this limitation in the revised version.

References

1. Palsson-McDermott EM, Curtis AM, Goel G, et al. Pyruvate kinase M2 regulates Hif-1 α activity and IL-1 β induction and is a critical determinant of the warburg effect in LPS-activated

macrophages. *Cell Metab.* 2015;2:65-80.

2. Chen X, Li YY, Zhang WQ, et al. House dust mite extract induces growth factor expression in nasal mucosa by activating the PI3K/Akt/HIF-1 α pathway. *Biochem Biophys Res Commun.* 2016;469:1055-61.

3. Differences in cell types

It is inconsistent because *Sirt6* is expressed in airway epithelial cells in mice, while in peripheral blood in human (Figure 1). Is *Sirt6* elevated in a cell-nonspecific manner? Different cells might be experiencing different phenomena?

R: We highly appreciate this valuable comment. Using published single-cell RNA-seq (scRNA-seq) data [*Nat Med.* 2019; 25:1153-1163], after quality control and first dimensionality reduction clustering, epithelial cells and other types of cells were extracted for secondary dimensionality reduction clustering. The analysis of database revealed that airway epithelial cells (Fig. 1A and B) and other cells such as macrophages, different T cell populations have an expression of SIRT6 (Fig. 1C and D), indicated that SIRT6 is expressed in a cell-nonspecific manner.

In fact, besides SIRT6 was elevated in airway epithelium cells in this study, we also investigated whether SIRT6 is elevated in macrophages (Unpublished data). We generated mice with a conditional *Sirt6* deletion in myeloid cells (mainly in macrophages, *Sirt6^{fl/fl}-LysMcre*). Representative genotyping results are shown in Fig. 2A. Conditional disruption of *Sirt6* was further confirmed by WB and RT-PCR analysis (Fig. 2B, C). We observed that airway inflammation, chemokine ligand (CXCL) 1, and CXCL2 were substantially decreased in HDM/LPS-exposed *Sirt6^{fl/fl}-LysMcre* mice compared with *Sirt6^{fl/fl}* mice (Control) exposed to HDM/LPS (Fig. 2D-I).

These data indicated that SIRT6 expression may be upregulated in some cells following exposure to allergen, which may trigger the same cellular event, such as inflammatory cytokine secretion, promoting airway inflammatory response in asthma. However, the mechanisms of SIRT6 in different cell types remain further investigation. We have discussed this limitation in the revised version.

Fig.1 SIRT6 expression in different types of cells from published scRNA-seq data

Fig. 2 Myeloid-specific SIRT6 deletion attenuated airway inflammatory in allergic asthma mice

4. Sirt6-expressing cells

Sirt6-expressing cells should be evaluated more comprehensively. For example, what about the expression of alveolar epithelial cells in mice? If they are expressed, it should be shown how expression is altered in your conditional knockout mice. Hematopoietic

cells only looked at macrophages. Which cells have elevated Sirt6 in humans? The authors may be missing the cells that are really expressing Sirt6.

R: We highly appreciate this valuable comment. As your suggestion, SIRT6 was determined in mouse alveolar epithelial cells by immunofluorescence staining. Although found that SIRT6 (red) was expressed in alveolar epithelial cells (green) of the lung (Fig. 1A), the expression of SIRT6 wasn't altered in the lung tissue of SIRT6 conditional knockout mice (Fig. 1B).

As discussed in question 3 above, SIRT6 expression was elevated in airway epithelial cells and macrophages in asthmatic mouse model. To determine whether SIRT6 is also elevated in airway epithelial cells and macrophages in asthmatic patients, we examined the colocalization of SIRT6 with SCGB1A1 (airway epithelium cell marker) or CD68 (macrophage marker) in bronchial biopsy specimens. Consistent with animal data, we found that SIRT6 expression in airway epithelium cells and macrophages was increased in patients with asthma compared with those seen in control subjects (Fig. 2A-D).

Collectively, SIRT6 expression is elevated in some cells exposed to allergen and promotes airway inflammatory response in asthma. However, the cellular and molecular mechanisms of SIRT6 within specific cell types remain largely unclear and need to be investigated further. We have discussed this limitation in the revised version. We really appreciate your rigorous work attitude.

Fig. 1 SIRT6 was expressed in mouse alveolar epithelial cells

Fig. 2 SIRT6 was elevated in airway epithelial cells and macrophages in asthmatic patients

5. Conditional knockout mice

In your mouse model, eosinophils are decreased in the conditional knockout mice. I don't think it is an effect of IL-17. Could Sirt6 have other important functions besides IL-17 induction? In particular, Sirt6 has been implicated in aging, and Sirt6 knockout epithelial cells due to aging in these mice?

R: Thank you very much for your sincere advices. It is well established that IL-17-induced neutrophil recruitment into airways is implicated in the pathogenesis of asthma. To address your valuable comment, we searched PubMed for papers on the subject of "IL-17", "Eosinophils", and "Asthma". A recent study has demonstrated that IL-17 receptor gene-deficient mice showed a reduced recruitment of not only neutrophils but also eosinophils into airways upon antigen challenge^[1]. IL-17-deficient or IL-17 receptor-deficient mice showed decreased in Th2 cytokine levels^[1,2]. IL-17 mRNA level was correlated positively with IL-5 mRNA levels in sputum of asthmatic patients^[3]. Inhibition of IL-17 activity with an anti-IL-17 Ab remarkably reduced antigen induced airway infiltration of eosinophils and Th2 cytokine levels in bronchoalveolar lavage fluids (BALF)^[4]. Consistent with these observations, our present data showed that SIRT6 deficiency inhibited IL-17A secretion and remarkably reduced airway infiltration of eosinophils, neutrophils, the levels of IL-5, IL-13

(eosinophilic inflammation) and KC/CXCL8, CXCL1, CXCL2 (neutrophilic inflammation) in asthmatic mice. Taken together, these findings indicate that IL-17 plays a crucial role in the pathogenesis of asthma, which promoting neutrophilic inflammation and Th2 cell-mediated eosinophilic inflammation.

SIRT6 has been implicated in aging. As your suggestion, we investigated whether the senescence-regulating function of SIRT6 plays a role in the severe asthma model constructed in this study using 6- to 8-week-old mice. We found that *AE-Sirt6*^{Δ/Δ} mice didn't show aging-associated gray hair, reduced hair density, and hair loss (Fig. A). Lung tissues from each group of mice were taken for senescence-associated key senescence markers p21, p16, and β-galactosidase (SA-β-Gal) staining^[5]. We found that the expression of p21 and p16, the number of positive SA-β-gal staining cells were slightly increased in the lung tissue of *AE-Sirt6*^{Δ/Δ} mice (Fig. B and C).

Collectively, these findings indicated that SIRT6 deficiency inhibited IL-17 expression, which contributing to reduce neutrophilic inflammation and Th2 cell-mediated eosinophilic inflammation. However, we didn't rule out the dysfunction of epithelial cells due to aging in *AE-Sirt6*^{Δ/Δ} and need to be investigated further. We have discussed this limitation in the revised version.

References

1. Schnyder-Candrian S, Togbe D, Couillin I, et al. Interleukin-17 is a negative regulator of established allergic asthma. *J Exp Med* 2006; 203:2715-2725.
2. Nakae S, Komiyama Y, Nambu A, et al. Antigen-specific T cell sensitization is impaired in IL-17-deficient mice, causing suppression of allergic cellular and humoral responses. *Immunity* 2002; 17:375-387.
3. Bullens DM, Truyen E, Coteur L, et al. IL-17 mRNA in sputum of asthmatic patients: linking T cell driven inflammation and granulocytic influx? *Respir Res* 2006; 7:135.
4. Park SJ, Lee KS, Kim SR, et al. Phosphoinositide 3-kinase δ inhibitor suppresses interleukin-17 expression in a murine asthma model. *Eur Respir J*. 2010; 36:1448-59.
5. Wu Z, Lu M, Liu D, et al. m6A epitranscriptomic regulation of tissue homeostasis during primate aging. *Nat Aging*. 2023;3:705-721.

6. OSS

Does OSS have an inhibitory effect in human cells? Is it really beneficial to suppress Sirt6 with OSS for patients with severe asthma since even IL-17 antibodies were ineffective?

R: We appreciate the valuable comment. As your suggestion, we tested whether OSS have an inhibitory effect in human bronchial epithelial (HBE) cells using RT-PCR. The data revealed that HDM/LPS exposure significantly induced inflammatory cytokines such as *Cxcl1*, *Cxcl2*, and *Il-17a* expression, but OSS decreased the *Cxcl1*, *Cxcl2*, and *Il-17a* expression.

As discussed in question one above, interleukin-17-targeted therapy such as Brodalumab maybe more beneficial in a specific Th17-high asthma subjects. It is unknown whether Brodalumab would have been more efficacious if patients had been selected for study using a phenotype-specific approach, that is, the presence of significant sputum neutrophilia or absence of Th2 biomarkers. Although suppress Sirt6 with OSS attenuated airway inflammation in a mouse model of severe asthma, it is still unknown whether OSS is effective in patients with severe asthma and the role of OSS for preventing severe asthma requires further investigation. We have discussed this limitation in the revised version.

Minor points

1. Allergen-induced HBE cells sound strange.

R: Thank for your suggestion. We have changed it to "HDM/LPS-induced HBE cells" in the revised version.

2. Figure 7E should be labeled.

R: We have added it in the revised version.

Reviewer: 2

Major comments

1. In figure 1F-G, show that *Sirt6* mRNA is increased peripheral blood of asthma patients and even higher in severe asthma patients. The main findings of this manuscript highlight that *Sirt6* specifically from bronchial epithelium increases IL-17A, so the authors could investigate the expression of *Sirt6* in bronchial brushes comparing mild-moderate asthma patients and severe asthma patients with non-asthmatic controls. Therefore, comparison of systemic increase of *Sirt6* in peripheral blood may not translate to increased *Sirt6* expression in bronchial epithelial cells.

R: We appreciate this valuable comment. As your suggestion, experiment was conducted to investigate the expression of *Sirt6* in bronchial brushes comparing mild-moderate asthma patients and severe asthma patients with non-asthmatic controls by immunofluorescence staining. We examined the colocalization of SIRT6 with SCGB1A1 (airway epithelium cell marker) in bronchial biopsy specimens. We found that SIRT6 expression in airway epithelium cells for patients with severe asthma was higher than those in mild-moderate asthma patients and those in non-asthmatic controls. We have added this finding in the revised version.

Fig. 1 SIRT6 was elevated in airway epithelial cells in asthmatic patients

2. In figure 2O, *Sirt6*⁺ cells in airway epithelium was compared in airway biopsies between asthma patients and controls. Here, the *Sirt6*⁺ cells were comparable between severe asthma patients and mild-moderate asthma patients, even though the numbers of

donors are not sufficient for this conclusion. The authors should elaborate the method of measuring Sirt6⁺ cells in airway epithelium in the figure legends or in the methods section. The y-axis shows ‘Sirt6⁺ cells in airway epithelium (% of total cells)’, if the total number of cells in the tissue section is different (as seen in Figure 2N), it may impact the analysis. The method should also include how the airway epithelium was distinguished from other cell types, for instance, airway smooth muscle cells. This also applies to Figure 3K, where IL-17A⁺ cells in airway epithelium is shown.

R: Thanks a lot for your suggestion. As your suggestion, we have performed new experiments and statistics using Double-labeling Immunofluorescence. According to previously described methods^[1,2], double-labeling immunofluorescence with SIRT6&SCGB1A1 (airway epithelium cell marker)/DAPI (cell nucleus), IL-17A&SCGB1A1/DAPI in the lung tissues were used to measure SIRT6⁺ or IL-17A⁺ cells in airway epithelium. All of the section samples used for the analysis were viewed using confocal microscopy (Olympus FV3000; Japan) at ×400 magnification and digitalization software. Co-localization was confirmed by z-series through the cell nucleus and 3D reconstruction that allows viewing of cells in the x–z and y–z direction (z-step, 1 μm). The percentages of SIRT6&SCGB1A1/DAPI, IL-17A&SCGB1A1/DAPI co-labelled cells in the total airway epithelium cell (SCGB1A1) was quantified by Image J software. An individual blinded to experimental conditions did all the counting. We have added new data and relevant description in the revised version.

References

1. Lian D, He D, Wu J, et al. Exogenous BDNF increases neurogenesis in the hippocampus in experimental *Streptococcus pneumoniae* meningitis. *J Neuroimmunol*. 2016;294:46-55.

2. Guzman-Marín R, Suntsova N, Methippara M, et al. Sleep deprivation suppresses neurogenesis in the adult hippocampus of rats. *Eur J Neurosci*. 2005;22:2111-6.

3. In Figure 1I, the authors correlate FEV₁ to Sirt6 mRNA expression in all donors including controls, mild-moderate and severe asthma patients. To correlate the Sirt6 mRNA expression the correlation must be done only using mild-moderate and severe asthma patients and exclude controls. An example of this analysis is in Figure 1K, showing correlation of ACQ scores with Sirt6 mRNA expression.

R: Thank for your suggestion. We have re-analyzed the data in the revised version.

4. The characteristics of asthma patients in the peripheral blood study has been shown in Table S1. The percentage of eosinophils are higher in severe asthma patients compared to mild-moderate and controls. IL-17A levels in severe asthma patients is linked with neutrophilic inflammation. Therefore, it is relevant to show, if percentage of neutrophils are higher in severe asthma patients compared to the other 2 groups. In addition, depicting correlation between Sirt6 mRNA levels and neutrophils percentage, would strengthen the link between IL-17A and neutrophilic inflammation.

R: We highly appreciate this valuable comment. We have re-analyzed the data and found that the percentage of neutrophils were higher in severe asthma patients compared to the other two groups. Moreover, the Sirt6 mRNA levels were positively correlated with neutrophils percentage. We have added it in the revised version.

5. In figure 3K, the authors demonstrate decrease of eosinophils, neutrophils and lymphocytes in BALF of *AE-Sirt6^{Δ/Δ}* mice treated with HDM and LPS. As this an allergic mouse model did the levels of IL-5, IL-13 (eosinophilic inflammation) and CXCL-8 (neutrophilic inflammation) also decrease in the BALF? Nonetheless, it is impressive that the knockout of *Sirt6* specifically in bronchial epithelium, leads to overall decrease in airway inflammation and remodelling.

R: Thanks a lot for your suggestion. As your suggestion, we assessed the levels of IL-5, IL-13, and CXCL-8 (also known as KC) in the BALF using ELISA. We found that the levels of IL-5, IL-13, and CXCL-8 were significantly decrease in the BALF of *AE-Sirt6^{Δ/Δ}* mice treated with HDM and LPS compared with control mice.

It is well established that IL-17-induced neutrophil recruitment into airways is implicated in the pathogenesis of asthma. A recent study has demonstrated that IL-17 receptor gene-deficient mice show a reduced recruitment of not only neutrophils but also eosinophils into airways upon antigen challenge^[1]. IL-17-deficient or IL-17 receptor-deficient mice show decreases in Th2 cytokine levels, which are associated with reduced airway hypersensitivity^[1,2]. IL-17 mRNA levels correlate positively with IL-5 mRNA levels in sputum from asthmatic patients^[3]. Consistent with these observations, our present data showed that SIRT6 deficiency inhibited IL-17 secretion and remarkably reduced airway infiltration of eosinophils, neutrophils, the levels of eosinophilic inflammation (IL-5, IL-13) and neutrophilic inflammation (KC/CXCL8, CXCL1, CXCL2) in asthmatic mice. Taken together, these findings indicated that SIRT6 deficiency inhibited IL-17 expression, which contributing to neutrophilic inflammation and Th2 cell-mediated eosinophilic inflammation. We have added and discussed it in the revised version.

6. It has been previously shown, that the major source IL-17A are the Th17 cells. Therefore, is IL-17A⁺ CD4⁺ T cells increased in asthmatic mice compared to control mice (in *Sirt6^{fl/fl}* mice)? In Figure 4E, IL-17A in the BALF is increased, the source of this IL-17A could also be from Th17 cells. Considering in Figure 4B, Th17 immune response is increased in asthmatic mice (as shown by RNA-seq of airway tissue), Th17 could also be a source of IL-17A, apart from bronchial epithelium. There are also, other cell types that produce IL-17A, for instance ILC3, can the authors comment on this.

R: Thanks a lot for your suggestion. The expression of IL-17A⁺CD4⁺ T cells in the lung tissues of mice was determined. We found that the expression of IL-17A⁺CD4⁺ T cells was increased in asthmatic mice compared to control mice. As your suggestion, these data suggested that apart from bronchial epithelium, other cell types that produce IL-17A (e.g., Th17 and ILC3) could also be a source of IL-17A. We have comment on it in the revised version.

7. In Figure 4Q, HDM in combination with LPS increase IL-17A production in in vitro culture HBE. Does HDM or LPS alone have the capacity to increase IL-17A production? Is the increased IL-17A produced by HBE a synergistic effect i.e. observed only in combination or does LPS or HDM alone increase IL-17A? LPS alone is a potent stimuli,

so is HDM required to see this increased IL-17A production? If not, then the in vitro effects observed in HBE does not pertain to HDM induced allergic airway inflammation.

R: We appreciate the constructive criticism and suggestions. Common indoor and outdoor environmental exposures can influence airway inflammation in asthma. The presence of endotoxin during allergen sensitization can lead to neutrophilic airway inflammation in mice. HDM is a complex mixture, containing mite allergens and microbial products derived from the membrane of colonizing Gram-negative bacteria and environmental contamination, and the level of LPS is correlated with the severity of asthma and decline in lung function^[1]. Moreover, co-exposure to HDM and LPS has been used to establish neutrophil-dominated airway inflammation models that simulate severe asthma^[2,3]. Accordingly, HDM in combination with LPS was used in our study.

As your suggestion, we detected IL-17A expression (as well as SIRT6) in human bronchial epithelial (HBE) cells after HDM or LPS stimulation. We found that HDM or LPS alone activated IL-17A expression in a dose-dependent manner (Fig. A and B). IL-17A production was slightly increase in HBE cells exposed to low dose LPS (0.125 $\mu\text{g/ml}$) alone, but HDM significantly potentiated low dose LPS-induced IL-17A expression (Fig. C). Accordingly, low dose LPS (0.125 $\mu\text{g/ml}$) plus HDM (50 $\mu\text{g/ml}$) were used in our study. These data indicated that the increased IL-17A was produced by HBE cells a synergistic effect. We have added this finding and discussed it in the revised version.

References

1. Daan de Boer J, Roelofs JJ, de Vos AF, et al. Lipopolysaccharide inhibits Th2 lung inflammation induced by house dust mite allergens in mice. *Am J Respir Cell Mol Biol.* 2013;48:382-9.

2. Krishnamoorthy N, Doua DN, Brüggemann TR, et al. Neutrophil cytoplasts induce TH17 differentiation and skew inflammation toward neutrophilia in severe asthma. *Sci Immunol*. 2018;3:eaa04747.

3. Wang L, Netto KG, Zhou L, et al. Single-cell transcriptomic analysis reveals the immune landscape of lung in steroid-resistant asthma exacerbation. *Proc Natl Acad Sci U S A*. 2021 Jan 12;118:e2005590118.

8. In Figure 7C, Sirt6 inhibitor reduced airway inflammation as shown by H&E staining. In BALF of these mice, were the percentage of eosinophils and neutrophils also decreased?

R: Thanks a lot for your suggestion. We found that total cells, the percentage of eosinophils and neutrophils were also decreased in BALF of these mice. We have added this finding in the revised version.

9. Severe asthma patients have eosinophilic inflammation or neutrophilic inflammation or mixed granulocytic (both eosinophilic and neutrophilic inflammation). The authors can add to the discussion on the effectiveness of the Sirt6 inhibitor as a therapeutic for severe asthma with predominant eosinophilic inflammation. Does inhibiting IL-17A be effective in reducing eosinophilic inflammation in severe asthma patients?

R: We appreciate the valuable comment. As your suggestion, we have added the discussion on the effectiveness of the Sirt6 inhibitor as a therapeutic for severe asthma with predominant eosinophilic inflammation.

It is well established that IL-17-induced neutrophil recruitment into airways is implicated in the pathogenesis of asthma. However, a recent study has demonstrated that IL-17 receptor gene-deficient mice show a reduced recruitment of not only neutrophils but also eosinophils into airways upon antigen challenge^[1]. IL-17-deficient or IL-17 receptor-deficient mice show decreases in Th2 cytokine levels, which are

associated with reduced airway hypersensitivity^[1,2]. IL-17 mRNA levels correlate positively with IL-5 mRNA levels in sputum from asthmatic patients^[3]. These findings indicated that IL-17A maybe also contributed to Th2 cell-mediated eosinophilic inflammation. However, clinical trials are needed to evaluate the effectiveness of inhibiting IL-17A in reducing eosinophilic inflammation in severe asthma patients. We have discussed this issues in the revised version.

References

1. Schnyder-Candrian S, Togbe D, Couillin I, et al. Interleukin-17 is a negative regulator of established allergic asthma. *J Exp Med* 2006; 203:2715-2725.
2. Nakae S, Komiyama Y, Nambu A, et al. Antigen-specific T cell sensitization is impaired in IL-17-deficient mice, causing suppression of allergic cellular and humoral responses. *Immunity* 2002; 17:375-387.
3. Bullens DM, Truyen E, Coteur L, et al. IL-17 mRNA in sputum of asthmatic patients: linking T cell driven inflammation and granulocytic influx? *Respir Res* 2006; 7:135.

Minor comments

1. In figure 1C, Sirt1, 3, 5, expression is decreased in HDM sensitized and challenged mice. Is this decrease also observed in asthma patients compared to controls in peripheral blood?

R: As your suggestion, Sirt1, 3, 5, expression was assessed in peripheral blood using RT-PCR. We found that Sirt1, 3, 5, expression was decreased in patients with asthma compared to controls. We have added this finding in the revised version.

Reviewer: 3

1. In Figure 1 animal model section, whether there is validation on the success of the animal modeling, such as HE, IHC, type II inflammatory factor determination, etc.

R: Thank you very much for your sincere advices. Severe asthma patients have eosinophilic inflammation or neutrophilic inflammation or mixed granulocytic (both eosinophilic and neutrophilic inflammation). Common indoor and outdoor environmental exposures can influence airway inflammation in asthma. The presence of endotoxin during allergen sensitization can lead to neutrophilic airway inflammation in mice. HDM is a complex mixture, containing mite allergens and microbial products. LPS is an endotoxin derived from the membrane of colonizing Gram-negative bacteria and environmental contamination, and the level of LPS is correlated with the severity of asthma and decline in lung function^[1]. Moreover, co-exposure to HDM and LPS has been used to establish neutrophil-dominated airway inflammation models that simulate severe asthma^[2-6]. Accordingly, a mouse model of severe asthma was developed by HDM and LPS.

We found that the total inflammatory cells, neutrophils, eosinophils, and lymphocytes in bronchoalveolar lavage fluid (BALF) were reduced in HDM/LPS-exposed *AE-Sirt6*^{Δ/Δ} mice (Fig. A and B). HE staining and quantitative analysis showed that SIRT6 deficiency attenuated HDM/LPS-induced peribronchial infiltrates of inflammatory cells (Fig. C and D). Several features of airway remodeling examined by peribronchial trichrome (Masson) staining, periodic acid Schiff (PAS) staining, and α -smooth muscle actin (α -SMA) were reduced in HDM/LPS-exposed *AE-Sirt6*^{Δ/Δ} mice (Fig. E and F). The levels of eosinophilic inflammation (IL-5, IL-13) and neutrophilic inflammation (KC/CXCL-8) were also reduced in HDM/LPS-exposed *AE-Sirt6*^{Δ/Δ} mice (Fig. G-I). Taken together, these findings indicated that there is validation on the success of the animal modeling.

2. In Figure 1, in the part of human experiment, whether there is protein level verification of PBMC to determine the change of sirt6, and whether there is mRNA and protein verification of human lung tissue.

R: We appreciate the valuable comment. We observed that SIRT6 protein level was significantly up-regulated in PBMC of patients with asthma compared with those seen in healthy controls (Fig. A). SIRT6 mRNA and protein expression were also increased in bronchial biopsies of asthmatic patients compared to controls (Fig. B, C). We have added this finding in the revised version.

3. Whether the enzymatic activity of sirt6 has changed in the mouse models and asthma patient.

R: We appreciate this valuable comment. As your suggestion, we only assessed the enzymatic activity of SIRT6 in patient with asthma because we didn't find the SIRT6

activity assay kit for mouse. We found that the enzymatic activity of SIRT6 was increased in asthmatic patients compared to controls.

4. In Figure 2 A, the expression of sirt6 seems to be inconsistent in the same lung tissue, is there any bias in the selection of view?

R: Thank you for this comment. To define the cellular source of SIRT6 in the lung tissue during asthma, we examined the colocalization of SIRT6 with SCGB1A1 (airway epithelium cells marker), F4/80 (macrophage marker), CD31 (endothelial marker), and α -SMA (smooth muscle cells marker) in the lung tissues of asthmatic mice using IF. We found that SIRT6 was largely expressed in airway epithelium cells, whereas low expression of SIRT6 in macrophages. However, we didn't detect the colocalization of SIRT6 with endothelial cells or smooth muscle cells. Therefore, the expression of SIRT6 was different in the different cell types and seemed to be inconsistent in the same lung tissue.

5. In lines 160 and 162, the figure labels do not match the picture labels.

R: Thank for your suggestion. We have corrected it in the revised version.

6. In Figure 5D, why not set a group which is only transfected with HA-SIRT6, not transfected with Flag-ROR γ t?

R: Thank for your suggestion. We have added it in the revised version.

7. In Figure 5K, the co-IP results show that the SIRT6-ΔC group still binds to RORγt, which is not consistent with the conclusion.

R: Thank for this comment. As your suggestion, the experiment was repeated. We have added it in the revised version.

8. Please label the antibodies used in the text, such as the brand and product number of the AcK antibody used for acetylation detection in figure 6.

R: Thank for your suggestion. We have added it in the revised version.

9. In Figure 6D, different sirt6 truncations affect the expression of IL17, whether there is WB protein level result?

R: Thank for this comment. As your suggestion, IL17 protein level result was added in the revised version.

10. In Figure6G, 6J, no input results are provided, such as IB Flag, Ace-α-tubulin, internal reference protein tubulin. In Figure6K, IB RORγt is missing in IP, and the result of internal reference protein is missing in input.

R: Thank for your suggestion. We have added it in the revised version.

Fig. 6G

Fig. 6J

Fig. 6K

REVIEWERS' COMMENTS

Reviewer #1 (Remarks to the Author):

The authors have sincerely investigated and answered the questions I raised.

Reviewer #2 (Remarks to the Author):

In this rebuttal, the authors have responded to all the major and minor comments in detail and have revised the manuscript accordingly. They have shown a novel role of SIRT6 in regulating IL-17A production in bronchial epithelium and SIRT6 inhibition reduces neutrophilic and eosinophilic inflammation and prevents airway remodelling.

A minor suggestion is to provide the line and page number of the changes made for tracking back the changes made with ease.

Reviewer #3 (Remarks to the Author):

Major concerns have been well addressed. The manuscript can be considered for acceptance if the following minor concerns are solved.

1. Figure 4 shows the RNA-SEQ results of AE-Sirt6 Δ/Δ mice modeled or not modeled for asthma, however, this paragraph focuses on the mechanism of Sirt6 in airway remodeling in asthmatic mice, whether it is better to use AE-Sirt6 fl/fl mice modeled as a control and AE-Sirt6 Δ/Δ mice modeled as an experimental group to perform RNA-SEQ.

The main subject of this paragraph should be sirt6, while sequencing using AE-Sirt6 Δ/Δ mice modeled or not, the variable seems to become asthmatic model.

2. Lines 231-235, the authors wanted to study whether IL17A in CD4+ T cells was affected by sirt6, but then below it became to study IL17A in airway epithelial cells, which needs to be explained, and also the cellular source or cell specificity of IL17A needs to be explained, because there is IL17A in more than just the airway epithelial cells, but the changes in sirt6 occur mainly in airway epithelial cells;

3. Sirt6 is a deacetylase that deacetylates ROR γ t and affects its acetylation level. The text shows that sirt6 also affects the protein expression level of ROR γ t, and what is its mechanism.

The point-by-point responses to the comments

Reviewer: 1

The authors have sincerely investigated and answered the questions I raised.

R: Thank you very much for your positive and constructive comments and suggestions, which have definitely improved the quality of our manuscript.

Reviewer: 2

In this rebuttal, the authors have responded to all the major and minor comments in detail and have revised the manuscript accordingly. They have shown a novel role of SIRT6 in regulating IL-17A production in bronchial epithelium and SIRT6 inhibition reduces neutrophilic and eosinophilic inflammation and prevents airway remodelling. A minor suggestion is to provide the line and page number of the changes made for tracking back the changes made with ease.

R: Thank you very much for your positive and constructive comments and suggestions, which have definitely improved the quality of our manuscript.

Reviewer: 3

1. Figure 4 shows the RNA-SEQ results of *AE-Sirt6^{Δ/Δ}* mice modeled or not modeled for asthma, however, this paragraph focuses on the mechanism of Sirt6 in airway remodeling in asthmatic mice, whether it is better to use *AE-Sirt6^{fl/fl}* mice modeled as a control and *AE-Sirt6^{Δ/Δ}* mice modeled as an experimental group to perform RNA-SEQ. The main subject of this paragraph should be *Sirt6*, while sequencing using *AE-Sirt6^{Δ/Δ}* mice modeled or not, the variable seems to become asthmatic model.

R: We appreciate the constructive criticism and suggestions.

We apologize for not writing it clearly in the article. In fact, we first performed RNA sequencing (RNA-seq) for the airway tissues from wild-type (WT) mice exposed to normal saline or HDM/LPS. We found that Th17-type immune response and IL-17A expression, which play a critical role in airway remodeling, were significantly upregulated in asthma mice compared with control mice. Further, we determined

whether SIRT6 regulates epithelial IL-17A expression in asthmatic airway remodeling using *Sirt6^{fl/fl}* mice and *AE-Sirt6^{Δ/Δ}* mice. We observed that HDM/LPS exposure increased the expression of IL-17A mRNA and protein in the airway tissues of *Sirt6^{fl/fl}* mice, whereas significantly decreased in *AE-Sirt6^{Δ/Δ}* mice (Fig. A-F), suggesting that SIRT6 can regulate the expression of IL-17A in asthma mouse model. However, as your suggestion, we all agree that it is better to use *AE-Sirt6^{fl/fl}* mice modeled as a control and *AE-Sirt6^{Δ/Δ}* mice modeled as an experimental group to perform RNA-SEQ. We have made detailed modifications to the result section in the revised version. We really appreciate your earnest work attitude and it is worthy of our learning.

2. Lines 231-235, the authors wanted to study whether IL-17A in CD4⁺ T cells was affected by sirt6, but then below it became to study IL-17A in airway epithelial cells, which needs to be explained, and also the cellular source or cell specificity of IL-17A needs to be explained, because there is IL-17A in more than just the airway epithelial cells, but the changes in sirt6 occur mainly in airway epithelial cells.

R: Thank you very much for your sincere advices. Actually, Lines 231-235, "To determine whether SIRT6 regulates IL-17A expression in severe asthma, we first assessed whether SIRT6 deficiency affects the IL-17A expression in CD4⁺ T cells", "CD4⁺ T cells" should be changed to "airway epithelial cells". We are very sorry for our incorrect writing. As your suggestion, although IL-17A expression was increased

in the airway epithelial cells in our study, it should be noted that other cell types such as Th17, ILC3 could also be a source of IL-17A. Therefore, the increased expression of IL-17A should be interpreted cautiously due to this limitation. We have comment on it in the revised version. Thanks again your rigorous work attitude.

3. *Sirt6* is a deacetylase that deacetylates ROR γ t and affects its acetylation level. The text shows that *sirt6* also affects the protein expression level of ROR γ t, and what is its mechanism.

R: We appreciate the valuable comment. The possible mechanisms are as follows. Protein stability: SIRT6 deacetylation can influence the stability of ROR γ t. Increased acetylation, as a result of *Sirt6* knockout, may make ROR γ t more susceptible to proteasomal degradation, resulting in lower protein levels. Acetylation levels: SIRT6 is known to deacetylate ROR γ t. When *Sirt6* is knocked out, the acetylation levels of ROR γ t may increase, which can lead to changes in its stability and activity. Increased acetylation can promote protein degradation or affect its DNA binding capacity, leading to changes in the targeted protein activity or stability¹⁻³. However, the underlying mechanism of SIRT6 affects the protein level of ROR γ t need to be further studied. We have discussed it in the revised version.

References

1. Yuan Z, Zhang X, Sengupta N, Lane WS, Seto E. Sirt1 regulates the function of the nijmegen breakage syndrome protein. *Mol Cell*. 2007;27:149-62
2. Lee JT, Gu W. Sirt1: Regulator of p53 deacetylation. *Genes Cancer*. 2013;4:112-7
3. Li Z, Zhu WG. Targeting histone deacetylases for cancer therapy: From molecular mechanisms to clinical implications. *Int J Biol Sci*. 2014;10:757-70